# RTDIFF: REVERSE TRAJECTORY SYNTHESIS VIA DIFFUSION FOR OFFLINE REINFORCEMENT LEARNING

**Qianlan Yang    Yu-Xiong Wang**
University of Illinois Urbana Champaign
{qianlan2,yxw}@illinois.edu

## ABSTRACT

In offline reinforcement learning (RL), managing the distribution shift between the learned policy and the static offline dataset is a persistent challenge that can result in overestimated values and suboptimal policies. Traditional offline RL methods address this by introducing conservative biases that limit exploration to well-understood regions, but they often overly restrict the agent's generalization capabilities. Recent work has sought to generate trajectories using generative models to augment the offline dataset, yet these methods still struggle with overestimating synthesized data, especially when out-of-distribution samples are produced. To overcome this issue, we propose RTDiff, a novel diffusion-based data augmentation technique that synthesizes trajectories *in reverse*, moving from unknown to known states. Such reverse generation naturally mitigates the risk of overestimation by ensuring that the agent avoids planning through unknown states. Additionally, reverse trajectory synthesis allows us to generate longer, more informative trajectories that take full advantage of diffusion models' generative strengths while ensuring reliability. We further enhance RTDiff by introducing flexible trajectory length control and improving the efficiency of the generation process through noise management. Our empirical results show that RTDiff significantly improves the performance of several state-of-the-art offline RL algorithms across diverse environments, achieving consistent and superior results by effectively overcoming distribution shift. Our code can be found at https://yanqval.github.io/RTDiff.

## 1 INTRODUCTION

Deep reinforcement learning (RL) has become a powerful tool for tackling complex challenges across a wide array of fields, including mastering board games (Silver et al., 2016), achieving superhuman performance in video games (Mnih et al., 2015), and improving continuous control in robotics (Lillicrap et al., 2016). The success of deep RL algorithms is primarily due to their ability to interact with and learn from extensive datasets generated through environmental interactions. However, in real-world applications, accumulating such a vast amount of exploratory data is often impractical and costly. In critical domains like healthcare (Gottesman et al., 2019) and autonomous driving (Yu et al., 2018), every interaction carries significant costs or risks, making online data collection infeasible and unsafe. Offline RL (Lange et al., 2012; Levine et al., 2020) offers a solution by training agents on pre-existing datasets, thereby avoiding the risks and costs of online data generation. Nevertheless, the transition from online to offline RL is challenging. Directly applying online RL techniques to offline RL tasks usually results in poor performance (Fujimoto et al., 2019; Wu et al., 2019). This is mainly due to the *distribution shift* between the policy derived from the offline data and the policy that originally generated the data. Such a shift can lead to overestimation of the unseen data in the offline dataset, producing inaccurate value estimates and suboptimal policies.

A typical solution is to develop advanced offline RL algorithms that incorporate a conservative bias into the learning process. These algorithms limit the policy search to regions within the offline dataset where there is high confidence. Model-free offline RL approaches, such as those proposed by Fujimoto et al. (2019) and Wu et al. (2019), embed this bias directly into their policy or value functions through conservative regularizations or specialized network architectures. Although these strategies effectively mitigate the problems associated with distribution shift, they may overly restrict the policy search, limiting the agent's ability to generalize beyond the specific confines of the offline

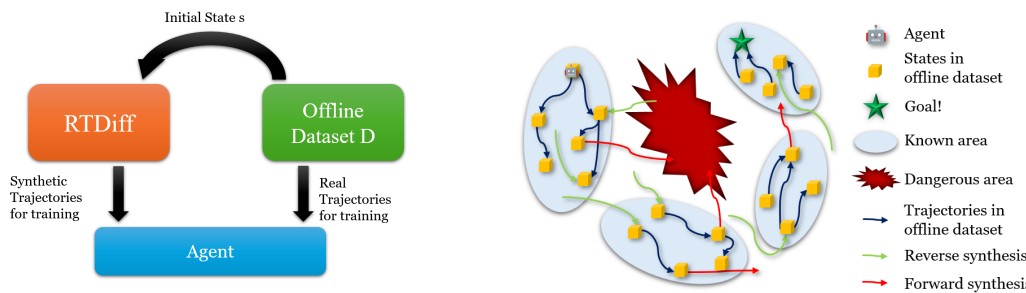

(a) The main paradigm of RTDiff.                    (b) Illustrative example.

Figure 1: **Illustration of RTDiff.** (a) RTDiff is used to augment the training dataset for reinforcement learning agents. It receives initial states $s$ from the offline dataset $D$ and synthesizes reverse trajectories. Both real and synthesized trajectories are utilized to train the RL agent. (b) An illustrative example is shown. In this figure, blue arrows represent trajectories inside the offline dataset, red arrows represent trajectories synthesized in the forward direction, and green arrows represent trajectories synthesized in the reverse direction. Forward trajectories include paths from known areas to dangerous areas, which might hinder performance. In contrast, reverse trajectories only include paths from dangerous areas to known areas, which do not adversely affect the agent.

dataset. Conversely, model-based offline RL methods (Kidambi & Rajeswaran, 2020; Yu et al., 2020; Lee et al., 2021; Yu et al., 2021) adopt a different strategy. These methods begin by learning a forward dynamics model that integrates conservative estimates from the offline dataset. They then use this model to generate imaginary trajectories, thereby expanding the dataset with high-confidence synthetic data.

Inspired by the promise of data synthesis, we turn our focus to state-of-the-art generative models. Diffusion models, known for their generative capabilities in computer vision and natural language processing, have been noticed by the RL researchers. Janner et al. (2022) and Ajay et al. (2023) introduce a foundational approach for decision-making in RL by generating full trajectories through a single denoising process. Building on the use of diffusion models in planning, Lu et al. (2023) propose synthesizing transitions using diffusion models trained on offline datasets to augment available data. More recent work (Yang & Wang, 2024; He et al., 2023) has expanded these ideas, applying diffusion models to generate trajectories that accelerate RL training. However, these approaches face the persistent challenge of distribution shift in data synthesis, which can lead to overestimation of values when out-of-distribution data are synthesized relative to the offline dataset.

To address this issue, in this paper, we propose a simple yet effective strategy overlooked by the community: **generating reverse trajectories instead of forward ones**. Our key insight lies in the intuition that overestimating a trajectory that begins from an *unknown* state and moves to a *known* state does not affect performance, as the agent will not pass through the unknown state during planning. Here, *known* states refer to states that are close to the distribution of states in the offline dataset, while *unknown* states are relatively far from this data distribution. During the planning phase, the agent always starts from a known state, where an overestimated trajectory originating from such states may risk leading the agent to *unknown* states, ultimately harming performance. However, by replacing such trajectories with those starting from *unknown* states and moving to *known* states, these risks are naturally reduced. Therefore, *reverse* synthesis directly addresses the issue of distribution shift. Additionally, by eliminating concerns about overestimating out-of-distribution trajectories, we can generate longer trajectories, providing the agent with more information and leveraging the ability of diffusion models to produce long, reliable trajectories.

Based on this insight, our work introduces a novel paradigm in offline RL by incorporating a diffusion model with reverse synthesis, termed Reverse Trajectory Diffuser (RTDiff). As illustrated in Fig. 1, our approach generates trajectories that *converge towards* target states already present in the offline dataset. By incorporating flexible generation length control, we can extend the trajectory length as much as possible to maximize the benefits of the diffusion model while ensuring the reliability of the generated data. Furthermore, by controlling the initial noise received by the diffusion model, we enhance the generation efficiency of our model.

**Our contributions** are three-fold. **(i)** We propose RTDiff, a novel diffusion-based approach that synthesizes reverse trajectories to address distribution shifts in offline reinforcement learning, which is general and can be integrated into a variety of offline reinforcement learning algorithms to improve their performance. **(ii)** We develop an out-of-distribution detector-based method that automatically adjusts the length of the generated trajectories. **(iii)** We reduce the number of generated samples by improving the generation efficiency through noise control. Empirical evaluation shows that RTDiff consistently achieves state-of-the-art performance across a variety of offline reinforcement learning environments.

## 2 RELATED WORK

**Offline RL.** In the realm of offline reinforcement learning, researchers focus on generalizing to out-of-distribution (OOD) data and avoiding the overestimation bias. These methods can be categorized into two main types: model-free and model-based algorithms. Model-free offline RL methods typically restrict their policy search to the offline dataset. This can be achieved through various ways, such as explicitly constraining the learning policy to remain close to the dataset (Fujimoto et al., 2019; Liu et al., 2020b), learning a conservative value function (Kumar et al., 2020), and applying importance sampling-based algorithms (Precup et al., 2001; Liu et al., 2020a). Additionally, they may estimate uncertainty quantification for the value function (Levine et al., 2020; Agarwal et al., 2020).

On the other hand, model-based offline RL methods have explored several different strategies. These include model-uncertainty quantification (Kidambi & Rajeswaran, 2020; Yu et al., 2020; Ovadia et al., 2019), representation learning (Lee et al., 2021), and constraining the policy to imitate the behavioral policy (Matsushima et al., 2020). They also use conservative estimation of the value function to enhance performance (Yu et al., 2021). Recently, new strategies have emerged for solving offline RL, such as treating RL problems as a sequence generation problem (Chen et al., 2021; Janner et al., 2021) or repurposing diffusion models for planning (Janner et al., 2022; Ajay et al., 2023).

**Diffusion Models in RL.** Diffusion models have demonstrated impressive capabilities in reinforcement learning, particularly in enhancing long-term planning and policy expressiveness. Prior work (Janner et al., 2022; Ajay et al., 2023) introduces a paradigm where diffusion models are used to construct full trajectories through conditioned sampling, guided by various criteria such as rewards, goal-oriented navigation, and skill deployment. These methods exploit the unique ability of diffusion models to generate extensive trajectories, effectively addressing challenges like long horizons and sparse rewards in RL. Other work (Du et al., 2023; He et al., 2023) extends this paradigm to generate visual-based data using diffusion models. Lu et al. (2023) employs a diffusion model to generate transitions that supplement the replay buffer, offering a distinctive strategy compared to earlier approaches. He et al. (2023) further positioned diffusion models as data synthesizers for generating trajectories to solve multi-task RL problems. More recently, ATraDiff (Yang & Wang, 2024) introduces a general framework that leverages offline data to generate full synthetic trajectories, improving the performance of online RL methods.

**Data Augmentation in RL.** Data augmentation has become a key technique for improving RL performance. Traditional methods (Yarats et al., 2021; Laskin et al., 2020; Sinha et al., 2021) incorporate various data augmentations like adding noise or random translation on observations for visual-based RL. Such approaches aim to help the agent to learn on multiple views of the same observation to improve the robustness. Some other works focus on generating synthetic data based on the models learned from the offline dataset (Yu et al., 2020; Wang et al., 2021). Recently, different from disturbing existing data points, researchers have focused on generating new data points from models learned from the original dataset and upsampling the original dataset. Representative work includes using diffusion models to generate data (Lu et al., 2023; Ajay et al., 2023; Du et al., 2023; He et al., 2023; Yang & Wang, 2024).

## 3 BACKGROUND

**Markov Decision Process.** In this paper, we explore sequential decision-making tasks that can be represented as a Markov Decision Process (MDP), denoted by $\mathcal{M} = \langle S, A, T, R, \gamma \rangle$. In this framework, $S$ denotes the set of states, $A$ denotes the set of actions, and $\gamma \in [0, 1)$ is the discount factor.

The functions $T(s'|s,a)$ and $R(s,a)$ describe the transition dynamics and reward structure, respectively. At each time step $t$, the agent selects an action $a \in A$, resulting in a new state $s'$ based on the transition function $T(s'|s,a)$ and receives an immediate reward $R(s,a)$. A trajectory in this context is a sequence of states and actions, expressed as $(s_1, a_1, r_1, s_2, a_2, r_2, \ldots, s_t, a_t, r_t)$, where $s_l$, $a_l$ and $r_l$ represent the state, action, and reward at time step $l$, respectively. Similarly, we define the reverse trajectory starting from the state $s_l$ is the sequence $(s_l, a_l, r_l, s_{l-1}, a_{l-1}, r_{l-1}, \ldots, s_1, a_1, r_1)$.

**Diffusion Models.** Diffusion probabilistic models conceptualize the process of generating data as an iterative denoising sequence, represented by $p_\theta(\tau^{i-1}|\tau^i)$. This sequence reverses a forward diffusion mechanism, $q(\tau^i|\tau^{i-1})$, which incrementally adds noise to the data over $N$ steps, thus degrading its structure. The resulting data distribution is described by:

$$p_\theta(x^0(\tau)) = \int p(x^N(\tau)) \prod_{i=1}^{N} p_\theta(x^{i-1}(\tau)|x^i(\tau)) d\tau^{1:N},$$

where $p(\tau^N)$ acts as a standard Gaussian prior, and $p(\tau^0)$ corresponds to the original noiseless data. The model parameters, $\theta$, are optimized by minimizing a variational bound on the negative log-likelihood of the denoising process: $\theta^* = \arg\min_\theta -\mathbb{E}_{\tau^0}[\log p_\theta(\tau^0)]$. This reverse process is typically modeled as a Gaussian distribution with fixed, timestep-specific covariances:

$$p_\theta(x^{i-1}(\tau)|x^i(\tau)) = \mathcal{N}(x^{i-1}(\tau)|\mu_\theta(x^i(\tau), i), \Sigma^i).$$

**Offline RL.** Offline RL is a type of reinforcement learning algorithm where the agent will be offered with an offline dataset $D = \{(s, a, r, s')\}$ and try to learn a policy $\pi_D$ from the offline dataset $D$. In offline RL, the agent will not be allowed to interact with the environment for online data collecting. The offline dataset is usually collected through multi-source policies.

## 4 METHOD

We now present our approach to augmenting the offline dataset by training our generative model RTDiff on the offline dataset to synthesize trajectories reversely. We begin by introducing how we design and train RTDiff (Sec. 4.1) and then decide the length of the generated trajectories to support flexible generation space (Sec. 4.2). Finally, we introduce how to further improve the generation efficiency by adjusting the input noise of the diffusion model (Sec. 4.3).

### 4.1 TRAJECTORY GENERATOR

To capture the trajectory data distribution of the offline dataset, we train a diffusion model to solve a conditional generation problem:

$$\max_\theta \mathbb{E}_{\tau \sim D}[\log p_\theta(x^0(\tau)|y(\tau))], \tag{1}$$

where $x^0(\tau)$ is the final generated reverse trajectory and $y(\tau)$ is the generation condition. In the experiments our paper focuses on, we use proprioceptive information as the state. So, the generated trajectories should be relatively low-dimensional. Specifically, we define the generated reverse trajectory at $t$-th diffusion step with length $L$ as the following 2D array:

$$x^t(\tau) = \begin{bmatrix} s_0^t & s_{-1}^t & s_{-2}^t & \cdots & s_{-L}^t \\ a_0^t & a_{-1}^t & a_{-2}^t & \cdots & a_{-L}^t \\ r_0^t & r_{-1}^t & r_{-2}^t & \cdots & r_{-L}^t \end{bmatrix}. \tag{2}$$

In Eqn. 2, the $i+1$-th column of $x^t(\tau)$ is a concatenation of state $s_{-i}$, action $a_{-i}$ and reward $r_{-i}$, in which index 0 means the current step, $-i$ means the $i$-th last step. The generation condition is set to be the initial state $y(\tau) = s_0$.

**Training.** To obtain the training dataset of the diffusion model, we first sample trajectories with fixed length $L$ from the offline dataset. For a trajectory with length $n$ in the offline dataset, we divide it into $n - L + 1$ trajectories with length $L$ so that the whole dataset becomes a larger dataset with fixed length $L$. For the training of the diffusion model with $N$ denoising steps, we use the following loss:

$$L(\theta) = \mathbb{E}_{t \sim u(N), \epsilon \sim \mathcal{N}(0,I)} \left[ ||\epsilon - \epsilon_\theta(x^t(\tau), y(\tau), t)||^2 \right], \tag{3}$$

---

**Algorithm 1** Augment Offline Dataset $D$ with RTDiff

---

**Require:** Offline dataset $D$, augmentation size $M$
 1: Initialize the synthesized dataset $D_s = \emptyset$
 2: Train RTDiff $p_\theta$ with the dataset $D$
 3: **while** $|D_s| < M$ **do**
 4:    Sample state $s_0 \in D$ from the offline dataset
 5:    Generate the trajectory $\tau = (s_0, a_0, r_0, s_{-1}, a_{-1}, r_{-1}, \ldots, s_{-L}, a_{-L}, r_{-L})$ with RTDiff
 6:    Cut the trajectory with the OOD detector $\tau = (s_0, a_0, r_0, s_{-1}, a_{-1}, r_{-1}, \ldots, s_{-l}, a_{-l}, r_{-l})$
 7:    Add all the transitions of the trajectory $\tau$ to the synthesized dataset $D_s = D_s \cup \tau$
 8: **end while**
 9: Combine the synthesized dataset to the original dataset $D = D \cup D_s$

---

where $u(N)$ is the uniform distribution on $\{1, 2, \ldots, N\}$.

**Architecture.** Since our experimental tasks focus on proprioceptive environments, the generation content is relatively low-dimensional compared with pixel-based generation, and we thus do not use a similar U-Net architecture as image works. We parameterize $\epsilon_\theta$ with an MLP with skip connections from the previous layer followed as (Lu et al., 2023) but increase the network width to 4096. For the sampling process, we use EDM (Karras et al., 2022) as the sampling method and we set the diffusion steps to 128.

**Deployment.** As stated in Algorithm 1, to augment the given offline dataset, we first train our RTDiff on the given offline dataset. Then we repeatedly sample a state $s_0$ as the initial state from the original dataset $D$, and synthesize a reverse trajectory from the initial state $s_0$. After that, we will cut the trajectory by an out-of-distribution (OOD) detector which will be introduced in the next section, and add all the transitions in the trajectory to the synthesized dataset. We repeat this process until the size of the synthesized dataset reaches our expected size $M$.

## 4.2 GENERATION LENGTH CONTROL

RTDiff is limited to generating fixed-length trajectories due to the inherent characteristics of diffusion models. Our investigation indicates that the quality of the augmented dataset is sensitive to the length of these trajectories. When the generated trajectories are too short, the agent gains limited benefit from the diffusion model's capacity to produce extended, reliable trajectories, thereby restricting RTDiff's performance enhancement. Conversely, if the generated trajectories are too long, they will go into out-of-distribution regions, causing hallucinations. An excess of these unrealistic generations can significantly decrease performance. For a deeper empirical analysis of the performance of different generation lengths, please refer to Sec. 5.2.

Therefore, we aim to design a flexible length control mechanism that can automatically adjust the generation length. This mechanism will ensure that each generated trajectory is as long as possible without excessively entering out-of-distribution regions.

We train an out-of-distribution detector $d(s)$ on the state space with the offline dataset, where it measures the distance of a state $s$ to the distribution of the given offline dataset. Ideally, we consider $d(s) > 1$ to represent that $s$ is an out-of-distribution state. To cut the generated trajectory $(s_0, a_0, r_0, s_{-1}, a_{-1}, r_{-1}, \ldots, s_{-L}, a_{-L}, r_{-L})$, we find the smallest $l + 1$ that $d(s_{-(l+1)}) > dis_M$, then we drop the trajectory after $s_{-l}$. Here $dis_M$ is a hyperparameter of our method, the details about how we choose this hyperparameter are included in Appendix C.3

**Out-of-Distribution Detector.** We train the OOD detector only by using the data points in the offline dataset. We design the OOD detector following the classic OOD work SSD (Sehwag et al., 2021). The main training scheme is: we first train a feature representation method to get the features of the states. Then, we partition the offline dataset into $m$ clusters based on the trained features, denoting each cluster as $Z_m$. Finally, the distance of a state is defined as $d'(s) = \min D_m(s, Z_m)$ where $D_m(,)$ is the Mahalanobis distance. We normalize the distance by dividing the maximum distance of states in the offline dataset $D$, $d(s) = \frac{d'(s)}{\max_{s' \in D} d'(s')}$ to make this threshold generalizable to all the environments.

### 4.3 IMPROVING GENERATION EFFICIENCY

Although RTDiff can enhance the performance of offline RL by augmenting the dataset, the number of generated samples can still be quite large, resulting in increased time and cost. To address this, we propose a technique aimed at improving the generation quality of RTDiff, thereby reducing the number of samples required in the data augmentation process and increasing sample efficiency.

Intuitively, if we generate two similar trajectories from the same initial state, the information carried by the two trajectories will be less than two independent trajectories. Therefore, we need to maximize the dissimilarity between all generated trajectories, which means reducing the correlation between the generated trajectories.

To this end, we propose a simple strategy that controls the noise $x^N(\tau)$ we choose for the diffusion model. Instead of separately generating the noises $x_1^N(\tau), x_2^N(\tau), \ldots, x_n^N(\tau)$ for one single initial state, we generate these noises together so that they will be evenly located within the space. That is we generate the noises by minimizing:

$$\min_{x_1^N(\tau), x_2^N(\tau), \ldots, x_n^N(\tau)} \sum_{i=1}^{n} \sum_{j=1}^{n} x_i^N(\tau) \cdot x_j^N(\tau), \tag{4}$$

which repulses noise vectors from each other.

Conceptually, this strategy is effective. As the diffusion model itself has the ability to generate diversified trajectories at a sufficiently high probability (this means the diffusion model will not always generate the same trajectory for all noises), we can assume that: for a trajectory $\tau_A$ generated by noise $x_A^N(\tau)$, with high probability, another trajectory $\tau_B$ generated from a randomly sampled noise $x_B^N(\tau) \sim N(0, 1)$ is sufficiently different from trajectory $\tau_A$, say $d(\tau_A, \tau_B) > d_0$, where $d_0$ is a constant. We can regard the full sampling process as a function from noise to trajectory. As the diffusion model's network and noise samplers are continuous, such sampling process is a continuous function. Using such continuity, we reach the conclusion that if we sample noise $x_B^N(\tau)$ only from the neighborhood of noise $x_A^N(\tau)$, with a high probability the generated trajectory $\tau_B$ from $x_B^N(\tau)$ will be close to trajectory $\tau_A$. Therefore, intuitively if we choose $x_B^N(\tau)$ which is located far from $x_A^N(\tau)$, we can avoid the neighborhood region (that fails at high probability) and have a higher probability to get a different trajectory.

## 5 EXPERIMENTS

We conduct a variety of experiments to verify the effectiveness of RTDiff. First, we validate that our approach can improve the performance of state-of-the-art offline RL algorithms in various environments (Sec. 5.1). Moreover, we conduct ablation studies to validate the effectiveness of different components in our approach (Sec. 5.2). Finally, we conduct additional experiments on visual RL tasks to show that our method can be extended to visual-based methods (Sec. 5.3). For evaluation, all results in this section are presented with the median performance over 5 random seeds.

### 5.1 MAIN RESULTS

We first show the overall performance of RTDiff, validating that RTDiff can improve the performance of state-of-the-art offline RL algorithms. We conduct experiments on 4 different environments from D4RL (Fu et al., 2020): Maze2d, AntMaze, Locomotion, and Kitchen. For comparison, we selected IQL (Kostrikov et al., 2022) and TD3+BC (Fujimoto & Gu, 2021) as benchmark offline RL algorithms, both widely regarded as state-of-the-art baselines. Additionally, we include experimental results for CQL (Kumar et al., 2020) and Decision Transformer (Chen et al., 2021), though these are omitted in the main paper due to space constraints; full results can be found in Appendix C.1. We select several data augmentation methods as baselines, including SynthER (Lu et al., 2023) and ATraDiff (Yang & Wang, 2024).

As illustrated by the results shown in Tables 1 and 2, our RTDiff consistently improves the performance of the offline RL algorithms in a wide range of environments. Meanwhile, our method outperforms the other data augmentation baselines including SynthER (Lu et al., 2023) and ATraD-

Table 1: Overall results of D4RL Maze, Antmaze, and Kitchen environments. **The number denotes the performance increase by the data augmentation method compared to the original result.** RTDiff consistently enhances the performance of offline reinforcement learning algorithms in all these environments. In particular, RTDiff significantly improves the performance in maze and kitchen environments.

| Environment | Data Type | IQL (Kostrikov et al., 2022) | | | TD3+BC (Fujimoto & Gu, 2021) | | |
|---|---|---|---|---|---|---|---|
| | | **RTDiff** | **SynthER** | **ATraDiff** | **RTDiff** | **SynthER** | **ATraDiff** |
| maze2d | umaze | $\mathbf{8.3^{\pm 3.5}}$ | $4.3^{\pm 4.1}$ | $5.6^{\pm 5.0}$ | $\mathbf{10.2^{\pm 4.7}}$ | $8.3^{\pm 4.3}$ | $9.3^{\pm 0.5}$ |
| | medium | $\mathbf{3.3^{\pm 2.7}}$ | $0.8^{\pm 3.2}$ | $2.1^{\pm 5.6}$ | $\mathbf{9.8^{\pm 2.5}}$ | $6.3^{\pm 2.5}$ | $7.4^{\pm 3.3}$ |
| | large | $\mathbf{14.3^{\pm 3.3}}$ | $11.4^{\pm 2.8}$ | $12.4^{\pm 4.8}$ | $\mathbf{7.7^{\pm 4.5}}$ | $4.8^{\pm 3.6}$ | $4.3^{\pm 3.7}$ |
| antmaze-umaze | fixed | $\mathbf{5.2^{\pm 3.3}}$ | $4.9^{\pm 3.7}$ | $4.4^{\pm 4.0}$ | $\mathbf{5.7^{\pm 3.5}}$ | $5.4^{\pm 3.8}$ | $2.5^{\pm 4.6}$ |
| | diverse | $4.3^{\pm 2.7}$ | $4.3^{\pm 3.1}$ | $\mathbf{4.7^{\pm 4.9}}$ | $\mathbf{4.2^{\pm 3.1}}$ | $3.9^{\pm 2.8}$ | $3.5^{\pm 4.4}$ |
| antmaze-medium | play | $\mathbf{7.9^{\pm 4.2}}$ | $7.5^{\pm 3.6}$ | $6.6^{\pm 3.5}$ | $8.3^{\pm 3.2}$ | $\mathbf{8.4^{\pm 2.7}}$ | $7.6^{\pm 5.5}$ |
| | diverse | $\mathbf{9.2^{\pm 3.8}}$ | $8.5^{\pm 3.6}$ | $8.8^{\pm 4.6}$ | $\mathbf{8.9^{\pm 2.4}}$ | $8.3^{\pm 3.6}$ | $8.7^{\pm 3.1}$ |
| antmaze-large | play | $\mathbf{6.5^{\pm 3.5}}$ | $5.4^{\pm 2.8}$ | $5.7^{\pm 4.0}$ | $\mathbf{5.4^{\pm 2.5}}$ | $4.8^{\pm 2.0}$ | $5.0^{\pm 4.5}$ |
| | diverse | $\mathbf{6.3^{\pm 3.4}}$ | $5.7^{\pm 2.5}$ | $4.4^{\pm 4.5}$ | $\mathbf{6.2^{\pm 5.5}}$ | $4.7^{\pm 6.2}$ | $4.9^{\pm 7.1}$ |
| kitchen | complete | $\mathbf{6.6^{\pm 7.4}}$ | $3.4^{\pm 8.3}$ | $5.4^{\pm 9.2}$ | $5.3^{\pm 7.2}$ | $3.6^{\pm 6.5}$ | $\mathbf{5.6^{\pm 7.5}}$ |
| | partial | $\mathbf{13.6^{\pm 6.3}}$ | $8.3^{\pm 7.2}$ | $11.3^{\pm 7.9}$ | $\mathbf{14.2^{\pm 7.8}}$ | $6.4^{\pm 6.8}$ | $12.3^{\pm 9.1}$ |
| | mixed | $\mathbf{11.3^{\pm 8.5}}$ | $6.2^{\pm 9.1}$ | $8.4^{\pm 9.4}$ | $\mathbf{10.3^{\pm 7.5}}$ | $7.2^{\pm 7.7}$ | $9.3^{\pm 8.6}$ |

Table 2: Overall results of D4RL Locomotion environment. **The number denotes the performance increased by the data augmentation method compared to the original result.** RTDiff improves the performance of various offline RL methods in different tasks, achieving the state of the art performance.

| Environment | Data Type | IQL (Kostrikov et al., 2022) | | | TD3+BC (Fujimoto & Gu, 2021) | | |
|---|---|---|---|---|---|---|---|
| | | **RTDiff** | **SynthER** | **ATraDiff** | **RTDiff** | **SynthER** | **ATraDiff** |
| walker2d | random | $0.8^{\pm 1.3}$ | $0.1^{\pm 1.9}$ | $0.3^{\pm 1.3}$ | $0.0^{\pm 0.2}$ | $0.1^{\pm 0.3}$ | $0.0^{\pm 0.3}$ |
| | mixed | $1.3^{\pm 2.3}$ | $0.7^{\pm 4.3}$ | $0.4^{\pm 5.1}$ | $4.3^{\pm 2.5}$ | $4.2^{\pm 2.6}$ | $3.7^{\pm 2.3}$ |
| | medium | $1.1^{\pm 4.7}$ | $0.7^{\pm 3.7}$ | $0.8^{\pm 3.6}$ | $3.1^{\pm 2.6}$ | $2.6^{\pm 2.2}$ | $1.9^{\pm 2.3}$ |
| | medexp | $0.2^{\pm 0.4}$ | $0.0^{\pm 0.4}$ | $0.4^{\pm 0.3}$ | $0.2^{\pm 0.2}$ | $0.3^{\pm 0.3}$ | $0.1^{\pm 0.1}$ |
| hopper | random | $1.2^{\pm 0.2}$ | $0.5^{\pm 0.3}$ | $0.6^{\pm 0.5}$ | $5.3^{\pm 0.4}$ | $3.6^{\pm 0.5}$ | $1.5^{\pm 0.8}$ |
| | mixed | $17.2^{\pm 1.7}$ | $18.4^{\pm 2.4}$ | $13.5^{\pm 5.9}$ | $7.4^{\pm 9.5}$ | $5.4^{\pm 4.9}$ | $3.6^{\pm 6.3}$ |
| | medium | $10.7^{\pm 6.0}$ | $9.8^{\pm 4.8}$ | $7.8^{\pm 5.4}$ | $8.2^{\pm 3.9}$ | $4.6^{\pm 7.3}$ | $2.5^{\pm 8.3}$ |
| | medexp | $3.6^{\pm 4.4}$ | $2.4^{\pm 5.2}$ | $3.8^{\pm 7.3}$ | $7.2^{\pm 1.7}$ | $5.8^{\pm 1.3}$ | $2.9^{\pm 6.7}$ |
| halfcheetah | random | $3.5^{\pm 1.2}$ | $2.1^{\pm 2.1}$ | $2.7^{\pm 2.3}$ | $1.6^{\pm 1.5}$ | $1.8^{\pm 2.0}$ | $1.2^{\pm 2.8}$ |
| | mixed | $4.2^{\pm 0.5}$ | $3.3^{\pm 0.4}$ | $3.8^{\pm 1.7}$ | $4.2^{\pm 1.2}$ | $3.7^{\pm 1.7}$ | $2.4^{\pm 1.9}$ |
| | medium | $2.4^{\pm 0.2}$ | $1.5^{\pm 0.3}$ | $0.9^{\pm 1.2}$ | $1.4^{\pm 0.1}$ | $1.2^{\pm 0.1}$ | $0.6^{\pm 0.3}$ |
| | medexp | $0.6^{\pm 0.2}$ | $0.2^{\pm 0.1}$ | $0.5^{\pm 0.3}$ | $0.5^{\pm 0.4}$ | $0.1^{\pm 0.3}$ | $0.0^{\pm 0.4}$ |

iff (Yang & Wang, 2024), which shows that RTDiff could generate data with the higher quality to enhance the offline dataset.

In particular, RTDiff significantly enhances the performance of offline RL algorithms in the Maze2D and Kitchen environments. We hypothesize two main reasons for this improvement. First, for tasks with long horizons, RTDiff leverages diffusion models to generate extended yet reliable trajectories, which are crucial for performance gains. Second, in environments where the state space is large relative to the states covered by the offline dataset—resulting in a sparsely populated offline dataset—RTDiff can more effectively utilize reverse synthesis to fill in the gaps, thus providing greater benefits.

## 5.2 ABLATION STUDIES

To verify the effect of different components in our RTDiff, we conduct several ablation studies.

**Is reverse synthesis important to the performance?** The key contribution of our work is to synthesize trajectories reversely. We argue that reverse synthesis is better than normal synthesis for data augmentation. To show the importance of reverse synthesis, we conduct an ablation study comparing our method with a normal synthesis algorithm. Here the method 'Normal' represents the algorithm that synthesizes trajectories in the *forward* order, with all other details the same with RT-Diff. And the method 'Short' represents the algorithm synthesizing trajectories in the forward order, but the generation length has been set to be 3, which we found to be the best choice in forward synthesis. The results shown in Table 3 illustrate that reverse synthesis significantly outperforms normal synthesis. Meanwhile, we found that if we perform normal synthesis, we cannot benefit from long trajectory generation, as the performance of synthesizing 'Short' trajectories outperforms synthesizing 'Normal' trajectories.

Table 3: Ablation study on reverse synthesis and normal synthesis with different generation lengths. The results indicate that reverse synthesis significantly outperforms normal synthesis. Additionally, increasing the generation length in normal synthesis leads to a decrease in performance.

| Environment | Data Type | IQL (Kostrikov et al., 2022) | | | TD3+BC (Fujimoto & Gu, 2021) | | |
|---|---|---|---|---|---|---|---|
| | | **RTDiff** | **Normal** | **Short** | **RTDiff** | **Normal** | **Short** |
| maze2d | umaze | $\mathbf{8.3^{\pm 3.5}}$ | $4.3^{\pm 4.0}$ | $5.8^{\pm 3.8}$ | $\mathbf{10.2^{\pm 4.7}}$ | $3.5^{\pm 4.1}$ | $8.5^{\pm 3.9}$ |
| | medium | $\mathbf{3.3^{\pm 2.7}}$ | $2.4^{\pm 3.2}$ | $5.9^{\pm 4.1}$ | $\mathbf{9.8^{\pm 2.5}}$ | $3.4^{\pm 3.0}$ | $6.1^{\pm 3.6}$ |
| | large | $\mathbf{14.3^{\pm 3.3}}$ | $3.2^{\pm 3.5}$ | $7.8^{\pm 4.0}$ | $\mathbf{7.7^{\pm 4.5}}$ | $2.4^{\pm 3.7}$ | $8.1^{\pm 4.2}$ |
| kitchen | complete | $\mathbf{6.6^{\pm 7.4}}$ | $1.5^{\pm 7.0}$ | $3.1^{\pm 7.2}$ | $\mathbf{5.3^{\pm 7.2}}$ | $0.8^{\pm 6.8}$ | $2.6^{\pm 6.9}$ |
| | partial | $\mathbf{13.6^{\pm 6.3}}$ | $3.0^{\pm 6.5}$ | $8.2^{\pm 6.8}$ | $\mathbf{14.2^{\pm 7.8}}$ | $3.5^{\pm 7.1}$ | $5.9^{\pm 7.4}$ |
| | mixed | $\mathbf{11.3^{\pm 8.5}}$ | $1.9^{\pm 8.0}$ | $5.4^{\pm 8.3}$ | $\mathbf{10.3^{\pm 7.5}}$ | $3.2^{\pm 7.7}$ | $7.7^{\pm 7.9}$ |

**Effect of the generation length.** The flexible generation length control is also an important component of our method. We found that the quality of the generated data is very sensitive to the length of the trajectories. Here we conduct an ablation study to show the performance of RTDiff with different fixed generation lengths. From the experimental results in Table 4, we can see that the performance of different generation lengths varies significantly and our generation length control strategy outperforms every fixed generation length setting.

**The effect of noise control.** We conduct an ablation study on the noise control component of RTDiff. We test RTDiff with and without the noise control part, using three different quantities of generated samples. The results, shown in Table 5, indicate that the noise control component enhances overall performance. Notably, as the number of generated samples decreases, the performance improvement due to noise control becomes more noticeable.

**Is RTDiff better than model-based methods?** We include more results of baselines including model-based RL methods MOPO (Yu et al., 2020) and model-based reverse imagination method ROMI (Wang et al., 2021). The results shown in Table 6 illustrate that RTDiff still outperforms those model-based baselines.

## 5.3 VISUAL REINFORCEMENT LEARNING

Finally, we demonstrate the applicability of RTDiff to visual reinforcement learning tasks, using Meta-World (Yu et al., 2019) as the benchmark. We selected tasks with varying difficulty levels, ranging from easy to hard. To adapt RTDiff for image-based inputs, we employed different architectures to generate visual trajectories. Building on the general pipeline from ATraDiff (Yang & Wang, 2024), which synthesizes pixel-based trajectories, we used Stable Diffusion to directly generate trajectory images. Actions and rewards were then predicted from these generated images using diffusion layers as the feature map. In addition, we modified the training images used in ATraDiff to synthesize reverse trajectories by training in reverse order. For evaluation, we applied offline RL algorithms, including CQL (Kumar et al., 2020), TD3+BC (Fujimoto & Gu, 2021), and IQL (Kostrikov et al., 2022). We directly compare RTDiff with ATraDiff for the Visual RL setting. The results, summarized in Table 7, demonstrate that RTDiff generalizes well to visual reinforcement learning, consistently enhancing the performance of offline RL methods.

Table 4: Ablation study on the effect of generation lengths. Our method significantly outperforms approaches using fixed generation lengths. We observed that as the generation length increases, performance improves as well. However, if the generation length becomes too long, performance begins to decline.

| Environment | Data Type | CQL (Kumar et al., 2020) | | | | |
|---|---|---|---|---|---|---|
| | | **RTDiff** | **L=1** | **L=3** | **L=5** | **L=10** |
| maze2d | umaze | $12.3^{\pm3.5}$ | $8.3^{\pm3.0}$ | $9.4^{\pm3.2}$ | $10.4^{\pm3.4}$ | $6.4^{\pm2.8}$ |
| | medium | $8.3^{\pm2.7}$ | $6.2^{\pm2.5}$ | $6.6^{\pm2.6}$ | $7.2^{\pm2.7}$ | $5.3^{\pm2.4}$ |
| | large | $11.3^{\pm3.3}$ | $8.3^{\pm3.0}$ | $9.2^{\pm3.1}$ | $10.4^{\pm3.2}$ | $7.2^{\pm2.9}$ |
| kitchen | complete | $6.6^{\pm7.4}$ | $3.8^{\pm6.5}$ | $4.5^{\pm6.8}$ | $5.8^{\pm7.1}$ | $4.3^{\pm6.3}$ |
| | partial | $13.6^{\pm6.3}$ | $9.4^{\pm5.5}$ | $10.3^{\pm5.8}$ | $11.4^{\pm6.0}$ | $10.5^{\pm5.3}$ |
| | mixed | $11.3^{\pm8.5}$ | $7.4^{\pm7.5}$ | $8.5^{\pm7.8}$ | $9.3^{\pm8.0}$ | $8.4^{\pm7.2}$ |

Table 5: Ablation study on the effect of noise control. The noise control component indeed improves the overall performance of RTDiff, especially when the number of generation samples is smaller.

| Environment | Noise Control | IQL (Kostrikov et al., 2022) | | | TD3+BC (Fujimoto & Gu, 2021) | | |
|---|---|---|---|---|---|---|---|
| | | **0.5M** | **1M** | **3M** | **0.5M** | **1M** | **3M** |
| maze2d-umaze | Yes | $4.1^{\pm1.2}$ | $6.7^{\pm2.0}$ | $8.3^{\pm3.5}$ | $7.2^{\pm2.1}$ | $10.3^{\pm3.0}$ | $10.2^{\pm4.7}$ |
| | No | $2.3^{\pm1.0}$ | $3.1^{\pm1.8}$ | $6.3^{\pm4.1}$ | $4.8^{\pm1.5}$ | $8.6^{\pm2.8}$ | $8.3^{\pm4.3}$ |
| maze2d-medium | Yes | $4.3^{\pm1.3}$ | $5.8^{\pm1.9}$ | $3.3^{\pm2.7}$ | $3.5^{\pm1.1}$ | $8.5^{\pm2.7}$ | $9.8^{\pm2.5}$ |
| | No | $1.4^{\pm1.0}$ | $2.9^{\pm1.8}$ | $2.8^{\pm3.2}$ | $3.4^{\pm1.0}$ | $6.1^{\pm2.0}$ | $6.3^{\pm2.5}$ |
| maze2d-large | Yes | $4.6^{\pm1.4}$ | $7.3^{\pm2.2}$ | $14.3^{\pm3.3}$ | $5.2^{\pm1.6}$ | $9.4^{\pm2.9}$ | $7.7^{\pm4.5}$ |
| | No | $3.4^{\pm1.1}$ | $6.9^{\pm2.1}$ | $11.4^{\pm2.8}$ | $4.3^{\pm1.3}$ | $8.6^{\pm2.6}$ | $4.8^{\pm3.6}$ |

Table 6: Overall results of D4RL Maze2D environments. **The number denotes the performance increase by the data augmentation method compared to the original result.**

| Environment | Data Type | IQL (Kostrikov et al., 2022) | | | TD3+BC (Fujimoto & Gu, 2021) | | |
|---|---|---|---|---|---|---|---|
| | | **RTDiff** | **MOPO** | **ROMI** | **RTDiff** | **MOPO** | **ROMI** |
| maze2d | umaze | $\mathbf{8.3^{\pm3.5}}$ | $5.1^{\pm2.8}$ | $5.4^{\pm2.4}$ | $\mathbf{10.2^{\pm4.7}}$ | $9.6^{\pm4.0}$ | $9.6^{\pm3.3}$ |
| | medium | $\mathbf{3.3^{\pm2.7}}$ | $1.7^{\pm1.9}$ | $2.1^{\pm1.8}$ | $\mathbf{9.8^{\pm2.5}}$ | $8.8^{\pm2.0}$ | $9.4^{\pm1.7}$ |
| | large | $\mathbf{14.3^{\pm3.3}}$ | $5.9^{\pm2.5}$ | $8.1^{\pm2.2}$ | $\mathbf{7.7^{\pm4.5}}$ | $2.6^{\pm3.8}$ | $3.5^{\pm3.1}$ |

Table 7: Results of the offline experiments on Meta-World (Yu et al., 2019). The number denotes the success rate of completing the task. RTDiff consistently improves the performance of various offline reinforcement learning algorithms in different tasks and outperforms ATraDiff.

| Task Name | TD3+BC | | | CQL | | | IQL | | |
|---|---|---|---|---|---|---|---|---|---|
| | Original | RTDiff | ATraDiff | Original | RTDiff | ATraDiff | Orignal | RTDiff | ATraDiff |
| Basketball | 0.13 | 0.21 | 0.16 | 0.02 | 0.05 | 0.04 | 0.16 | 0.24 | 0.18 |
| Box Close | 0.03 | 0.07 | 0.04 | 0.00 | 0.01 | 0.00 | 0.08 | 0.12 | 0.09 |
| Push Wall | 0.06 | 0.12 | 0.08 | 0.02 | 0.03 | 0.02 | 0.09 | 0.14 | 0.12 |
| Coffee Push | 0.53 | 0.62 | 0.54 | 0.46 | 0.58 | 0.51 | 0.61 | 0.72 | 0.63 |
| Sweep | 0.19 | 0.24 | 0.22 | 0.16 | 0.21 | 0.17 | 0.24 | 0.26 | 0.24 |

# 6 VISUALIZATION AND ANALYSIS

To better understand why RTDiff improves the performance of offline reinforcement learning algorithms and why reverse synthesis avoids issues present in normal synthesis, we conduct an analysis using an illustrative environment. We design a task similar to Maze2D but simpler and cleaner for better analysis. In this task, the agent starts from the bottom-left corner and aims to reach the target in the top-left corner of the map. The agent receives a reward of 1 for approaching the goal. The left part of the map contains a dangerous area; if the agent enters this area, the episode ends immediately, and the agent receives a reward of $-10$. The offline dataset does not contain a full trajectory from the start to the end. Instead, there is a small area in the middle of the map that is not covered by any trajectories in the dataset.

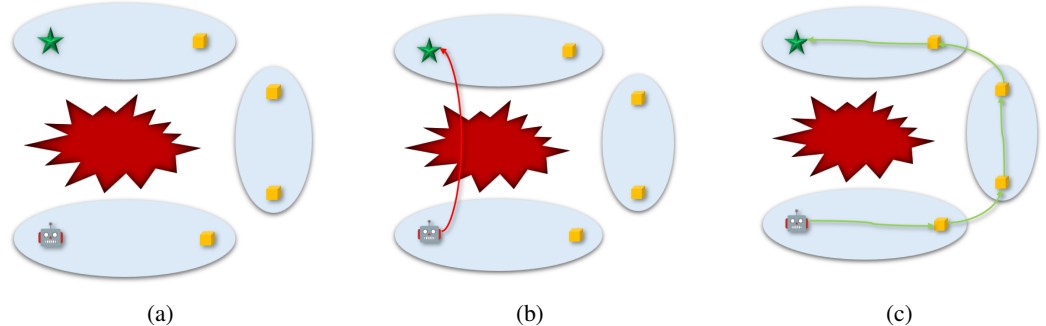

| (a) | (b) | (c) |

Figure 2: **Left**: The visualization of the illustrative environment. **Middle**: The trajectory of the offline RL agent with *normal* synthesis. **Right**: The trajectory of the offline RL agent with *reverse* synthesis. *Normal* synthesis will lead to a trajectory going across the dangerous area, while *Reverse* synthesize can help the agent to remain in the known area and reach the goal.

We test both reverse synthesis and normal synthesis in this environment. As illustrated in Fig. 3, normal synthesis generates trajectories that start from the lower area and enter the middle dangerous area. If these hallucinated trajectories are assigned relatively higher rewards, they can mislead the agent into the dangerous area, ultimately resulting in failed ex-

|                  | In2Out | Out2In | In2In | Out2Out |
|------------------|--------|--------|-------|---------|
| Reverse          | 2.7%   | 21.6%  | 48.0% | 27.3%   |
| Normal (w/o OOD) | 11.2%  | 5.8%   | 71.8% | 11.2%   |
| Normal (w/ OOD)  | 18.2%  | 6.2%   | 52.1% | 23.5%   |

Table 8: Ratios of different types of transitions generated by reverse synthesis and normal synthesis on the Maze2D-large environment. Normal synthesis generates much more **In2Out** transitions than reverse synthesis.

ecution. Conversely, reverse synthesis generates trajectories that move from the dangerous area to the upper or lower areas. Although these trajectories are also hallucinations, they do not impact the decision-making process because the agent will never make decisions from within the dangerous area. Consequently, the agent can successfully navigate to the goal location.

To further analyze this phenomenon in larger environments, we evaluated the ratio of four types of transitions: **In2In, Out2Out, In2Out**, and **Out2In**. In particular, **In2Out** transitions, which move from an in-distribution state to an out-of-distribution (OOD) state, are especially risky, as they may mislead the agent and degrade performance. In contrast, **Out2In** transitions, where the agent moves back from an OOD state to an in-distribution state, are less risky. We measure these ratios in the Maze2D-large environment for both reverse and normal synthesis methods, with and without an OOD detector. As shown in Table 8, normal synthesis produces significantly more **In2Out** transitions compared to reverse synthesis, even when an OOD detector is applied. This finding underscores the robustness of reverse synthesis in avoiding risky transitions, ultimately improving the performance of RL agents.

## 7 CONCLUSION

We introduce RTDiff, a diffusion-based offline reinforcement learning data augmentation method that synthesizes reverse trajectories. By incorporating the diffusion models to generate trajectories, we reversely generate long trajectories to augment the dataset. By using an out-of-distribution detector, we automatically adapt the length of the generated trajectories, improving the generation quality. By controlling the noise of the diffusion model, we remove redundant generations and improve the efficiency of generation. We test RTDiff in various environments and found that RTDiff consistently improves offline reinforcement learning algorithms, especially in some long-horizon, complicated environments.

**Limitation and Future Work.** Our work introduces a novel reinforcement learning data augmentation method that mitigates the issue of distributional shifts. However, our method still necessitates generating a large number of samples to achieve optimal performance, which results in extended inference running time. In future work, we aim to enhance the noise control component and improve sample efficiency, enabling us to achieve comparable results with significantly fewer samples.

ACKNOWLEDGMENTS

This work was supported in part by NSF Grant 2106825, NIFA Award 2020-67021-32799, the Toyota Research Institute, the IBM-Illinois Discovery Accelerator Institute, the Amazon-Illinois Center on AI for Interactive Conversational Experiences, Snap Inc., and the Jump ARCHES endowment through the Health Care Engineering Systems Center at Illinois and the OSF Foundation. This work used computational resources, including the NCSA Delta and DeltaAI supercomputers through allocations CIS220014 and CIS230012 from the Advanced Cyberinfrastructure Coordination Ecosystem: Services & Support (ACCESS) program, as well as the TACC Frontera supercomputer and Amazon Web Services (AWS) through the National Artificial Intelligence Research Resource (NAIRR) Pilot.

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

# Appendices

## A  ENVIRONMENTS TESTED

Following are the environments we evaluated in Sec. 5:

**D4RL Maze2d** (Fu et al., 2020). The maze2d task is a navigation task that requires a 2D agent to reach a fixed goal location in the maze. This task jusitifies the ability of offline RL algorithms to stitch previously collected subtrajectories to get the shortest path to the goal location. There are three layouts in this task, including umaze, medium and large. The dataset of this environment is generated by selecting waypoints randomly and using a planner which could generate subtrajectories among the waypoints.

**D4RL AntMaze** (Fu et al., 2020). The Antmaze task is a navigation task that replaces the 2D ball from Maze2D with a 8-Dof Ant quadraped robot. This task combines the challenges of controlling the robot and naviguting the robot to the goal location. There are three different layouts in this environment, including umaze, medium, and large. The environment also contains3 three flavors of datasets, including fixed, diverse, and play, wich differs in the chosen of the start and goal locations.

**D4RL Locomotion** (Fu et al., 2020). The Locomotion environment contains three different types of tasks (walker2d, hopper, and halfcheetah), including 12 different offline data with varying levels of expertise (random, medium, medium-replay, and medium-expert). The medium datasets are generated by a policy trained with a early-stopping SAC (Haarnoja et al., 2018). The random datasets are generated by a random initilized policy. The medium-replay datasets consist of samples in the replay buffer during the training until the policy reaches the medium performance. The medium-expert dataset contains part of the expert demonstrations and part of the suboptimal trajectories.

**D4RL Kitchen** (Fu et al., 2020). The Kitchen task involves a simulated environment where a 9-DoF robot manipulates various objects, such as sliding a cabinet door, switching an overhead light, and opening a microwave. Initially introduced by (Gupta et al., 2019), this task requires the robot to complete a sequence of multiple subtasks, each rewarded with a sparse, binary reward upon successful completion. The offline dataset provided includes only portions of the complete sequence, necessitating that the agent learn to assemble these sub-trajectories effectively.

**Meta-World** (Yu et al., 2019). Meta-World is an extensive platform created to assess and enhance algorithms in both reinforcement learning and multi-task learning. With 50 unique robotic manipulation tasks, it provides a varied and demanding setting for evaluating how well algorithms can generalize and rapidly learn new skills.

## B  HYPERPARAMETERS

We list all the hyperparameters here, which are applied to all the environments. In addition, we will release our code upon acceptance.

| Hyperparameter | Value |
|---|---|
| Batch Size | 16 |
| Training Steps | $10^6$ |
| Optimizer | Adam |
| Learning Rate | $2 \times 10^{-4}$ |
| Trajectory Length | 10 |
| Distance Threshold | 1.5 |
| Diffusion Steps | 128 |
| Number of Generations | $5 \times 10^6$ |

Table 9: Hyperparameter settings used in our experiments.

Table 10: Results of CQL and Decision Transformer on the D4RL Maze, Antmaze, and Kitchen environments. **The numbers denote the performance increase by the data augmentation method compared to the original result.** RTDiff consistently improves the performance of offline reinforcement learning algorithms in all these environments.

| Environment | Data Type | CQL (Kumar et al., 2020) | | | DT (Chen et al., 2021) | | |
|---|---|---|---|---|---|---|---|
| | | RTDiff | SynthER | ATraDiff | RTDiff | SynthER | ATraDiff |
| maze2d | umaze | $12.3^{\pm3.5}$ | $6.3^{\pm4.1}$ | $7.1^{\pm4.0}$ | $17.2^{\pm4.7}$ | $8.3^{\pm4.3}$ | $9.0^{\pm4.1}$ |
| | medium | $8.3^{\pm2.7}$ | $5.8^{\pm3.2}$ | $6.2^{\pm3.1}$ | $9.8^{\pm2.5}$ | $6.3^{\pm2.5}$ | $5.9^{\pm2.6}$ |
| | large | $11.3^{\pm3.3}$ | $7.4^{\pm2.8}$ | $7.8^{\pm2.9}$ | $12.7^{\pm4.5}$ | $7.8^{\pm3.6}$ | $7.5^{\pm3.7}$ |
| antmaze-umaze | fixed | $5.2^{\pm3.3}$ | $4.9^{\pm3.7}$ | $4.8^{\pm3.6}$ | $5.7^{\pm3.5}$ | $5.4^{\pm3.8}$ | $5.5^{\pm3.7}$ |
| | diverse | $4.3^{\pm2.7}$ | $4.3^{\pm3.1}$ | $4.3^{\pm3.0}$ | $4.2^{\pm3.1}$ | $3.9^{\pm2.8}$ | $4.0^{\pm2.9}$ |
| antmaze-medium | play | $7.9^{\pm4.2}$ | $7.5^{\pm3.6}$ | $7.4^{\pm3.5}$ | $8.3^{\pm3.2}$ | $8.4^{\pm2.7}$ | $8.2^{\pm2.8}$ |
| | diverse | $9.2^{\pm3.8}$ | $8.5^{\pm3.6}$ | $8.8^{\pm3.7}$ | $8.9^{\pm2.4}$ | $8.3^{\pm3.6}$ | $8.5^{\pm3.4}$ |
| antmaze-large | play | $6.5^{\pm3.5}$ | $5.4^{\pm2.8}$ | $5.6^{\pm3.0}$ | $5.4^{\pm2.5}$ | $4.8^{\pm2.0}$ | $4.6^{\pm2.1}$ |
| | diverse | $6.3^{\pm3.4}$ | $5.7^{\pm2.5}$ | $5.9^{\pm2.7}$ | $5.8^{\pm5.5}$ | $4.7^{\pm6.2}$ | $5.0^{\pm6.0}$ |
| kitchen | complete | $6.6^{\pm7.4}$ | $3.4^{\pm8.3}$ | $4.0^{\pm8.0}$ | $5.3^{\pm7.2}$ | $3.6^{\pm6.5}$ | $4.2^{\pm6.7}$ |
| | partial | $13.6^{\pm6.3}$ | $8.3^{\pm7.2}$ | $9.0^{\pm7.0}$ | $14.2^{\pm7.8}$ | $6.4^{\pm6.8}$ | $7.0^{\pm6.9}$ |
| | mixed | $11.3^{\pm8.5}$ | $6.2^{\pm9.1}$ | $6.0^{\pm9.0}$ | $10.3^{\pm7.5}$ | $7.2^{\pm7.7}$ | $7.5^{\pm7.6}$ |

Table 11: Results of CQL and DT on the D4RL Locomotion environment. **The numbers denote the performance increase by the data augmentation method compared to the original result.** RTDiff improves the performance of these reinforcement learning methods in different tasks.

| Environment | Data Type | CQL (Kumar et al., 2020) | | | DT (Chen et al., 2021) | | |
|---|---|---|---|---|---|---|---|
| | | RTDiff | SynthER | ATraDiff | RTDiff | SynthER | ATraDiff |
| walker2d | mixed | $5.2^{\pm2.3}$ | $4.9^{\pm4.3}$ | $5.1^{\pm3.8}$ | $2.2^{\pm1.3}$ | $2.4^{\pm2.4}$ | $2.2^{\pm2.0}$ |
| | medium | $2.6^{\pm4.7}$ | $2.3^{\pm3.7}$ | $2.5^{\pm4.1}$ | $2.3^{\pm2.1}$ | $2.1^{\pm2.8}$ | $2.2^{\pm2.5}$ |
| | medexp | $0.1^{\pm0.4}$ | $0.0^{\pm0.4}$ | $0.1^{\pm0.4}$ | $0.6^{\pm0.8}$ | $0.4^{\pm0.7}$ | $0.5^{\pm0.7}$ |
| hopper | mixed | $16.4^{\pm1.7}$ | $18.4^{\pm2.4}$ | $17.6^{\pm2.1}$ | $11.2^{\pm5.3}$ | $13.6^{\pm4.7}$ | $13.2^{\pm4.5}$ |
| | medium | $6.3^{\pm6.0}$ | $5.8^{\pm4.8}$ | $6.1^{\pm5.4}$ | $4.3^{\pm1.5}$ | $3.5^{\pm2.3}$ | $4.0^{\pm2.0}$ |
| | medexp | $5.3^{\pm4.4}$ | $3.6^{\pm5.2}$ | $4.9^{\pm4.8}$ | $1.6^{\pm1.2}$ | $1.3^{\pm2.2}$ | $1.5^{\pm1.9}$ |
| halfcheetah | mixed | $2.4^{\pm0.8}$ | $1.9^{\pm0.5}$ | $2.3^{\pm0.6}$ | $2.4^{\pm0.8}$ | $1.9^{\pm0.5}$ | $2.3^{\pm0.6}$ |
| | medium | $0.9^{\pm0.3}$ | $0.6^{\pm0.4}$ | $0.8^{\pm0.4}$ | $0.9^{\pm0.3}$ | $0.6^{\pm0.4}$ | $0.8^{\pm0.4}$ |
| | medexp | $1.3^{\pm0.8}$ | $0.0^{\pm0.6}$ | $1.0^{\pm0.7}$ | $1.3^{\pm0.8}$ | $0.0^{\pm0.6}$ | $1.0^{\pm0.7}$ |

## C    ADDITIONAL EXPERIMENTAL RESULTS

In this section, we show more experimental results to support the conclusion of our paper.

### C.1    RESULTS WITH DIFFERENT BASIC RL ALGORITHMS

To illustrate that our RTDiff indeed improves the performance of general offline RL methods, here we include more experiments involving Decision Transformer (Chen et al., 2021) and CQL (Kumar et al., 2020), which are representative sequence modeling baseline and model-free baseline. The results shown in Tables 10 and 11 illustrate that our method consistently improves the performance of different offline RL methods.

### C.2    ORIGINAL PERFORMANCE REPORT

The performance increase reported in Section 5.1 is measured by the difference between the normalized score with data augmentation and the original normalized score without any data augmentation methods. The original results are shown in Table 12.

Table 12: Original normalized return of the methods we used in our paper on the D4RL Locomotion environment.

| Environment | Data Type | CQL | TD3+BC | DT | IQL |
|---|---|---|---|---|---|
| walker2d | mixed | $73.1 \pm 13.2$ | $85.6 \pm 4.0$ | $81.8 \pm 6.9$ | $82.2 \pm 3.0$ |
| | medium | $80.8 \pm 3.3$ | $82.7 \pm 4.8$ | $65.1 \pm 1.6$ | $80.9 \pm 3.2$ |
| | medexp | $109.6 \pm 0.4$ | $110.0 \pm 0.4$ | $110.4 \pm 0.3$ | $111.7 \pm 0.9$ |
| hopper | mixed | $95.1 \pm 5.3$ | $64.4 \pm 21.5$ | $59.9 \pm 2.7$ | $97.4 \pm 6.4$ |
| | medium | $59.1 \pm 3.8$ | $60.4 \pm 3.5$ | $67.6 \pm 2.5$ | $67.5 \pm 3.8$ |
| | medexp | $95.1 \pm 5.3$ | $101.2 \pm 9.1$ | $107.1 \pm 1.0$ | $107.4 \pm 7.8$ |
| halfcheetah | mixed | $45.0 \pm 0.3$ | $44.8 \pm 0.6$ | $38.9 \pm 0.5$ | $44.5 \pm 0.2$ |
| | medium | $47.0 \pm 0.2$ | $48.1 \pm 0.2$ | $42.2 \pm 0.3$ | $48.3 \pm 0.2$ |
| | medexp | $95.6 \pm 0.4$ | $90.8 \pm 6.0$ | $91.6 \pm 1.0$ | $94.7 \pm 0.5$ |

### C.3  MORE ABLATION STUDIES

**Threshold of the OOD detector.** We select the value of this threshold with the following method, using D4RL Locomotion environment as the representative environment: We use grid search to find the best choice of the hyperparameter, and then do a cross-validation of the representative environment to ensure its robustness. After selecting the threshold, we directly apply this threshold to all the environments we used, without any further tuning. To demonstrate the robustness of our threshold, we conduct a further ablation study on the environment maze2d. The results shown in Table 13 illustrate that this threshold $dis_M = 1.5$ is reasonable across different environments.

Table 13: Performance of maze2d environments under different thresholds. $dis_M = 1.5$ achieves the overall best performance compared with other threshold choices.

| Threshold | 1.0 | 1.3 | 1.5 | 2.0 |
|---|---|---|---|---|
| maze2d-umaze | 7.2 | 12.5 | 12.3 | 4.1 |
| maze2d-medium | 5.7 | 7.9 | 8.3 | 2.8 |
| maze2d-large | 7.1 | 9.6 | 11.3 | 3.7 |

### C.4  RESULTS OF OTHER BASELINES WITH OOD DETECTOR AND NOISE CONTROL

To verify the effectiveness of different components in RTDiff, we incorporate them with other baselines to show the results.

Specifically, in our method, the OOD detector is used to control the length of the generated trajectories. This is crucial because we aim to generate trajectories that extend beyond the offline data distribution to provide new information to the agent, while ensuring they do not deviate too far, which could reduce their usefulness and increase risk. The effectiveness of our OOD detector is demonstrated in Tables 4 and 13. Our results show that the OOD detector outperforms any fixed-length generation strategy. Additionally, we observed that setting the threshold too high or too low negatively impacts performance.

To better understand why reverse synthesis is useful, we want to further show that this OOD detector is not useful for other synthesis methods. First of all, SynthER only performs transition-level synthesis, which inherently does not need to control the generation length. Also we want to emphasize that with forward synthesis like ATraDiff, generating trajectories going out of the offline data distribution is more risky, as transitions going from inside to outside may lead to performance degradation. To validate this, we show the results of combining ATraDiff with an OOD detector of different thresholds. From the results shown in Table 14, we found that those forward synthesis methods derive limited benefits from the OOD detector, which validated the unique effectiveness of reverse synthesis in our framework.

Our proposed noise control is largely independent of the specific generation method and can be applied to any data augmentation approach using a diffusion-based framework. It demonstrates particularly effective when the number of examples is limited. Notably, this technique is also applicable to SynthER and ATraDiff, as demonstrated by the results shown in Table 15. We use SynthER and

ATraDiff to generate both 1M data with and without the random generation.The results in Table 15 illustrate that the noise control can consistently improve the performance.

Table 14: Performance of ATraDiff with the OOD detector across different thresholds in the Maze2D environments. Forward synthesis cannot benefit from the OOD detector.

| Method | RTDiff | ATraDiff | $dis_M = 1.0$ | $dis_M = 1.3$ | $dis_M = 1.5$ | $dis_M = 2.0$ |
|---|---|---|---|---|---|---|
| maze2d-umaze | 12.3 | 7.1 | 7.4 | 7.0 | 6.3 | 6.0 |
| maze2d-medium | 8.3 | 6.2 | 6.6 | 6.3 | 5.6 | 5.4 |
| maze2d-large | 11.3 | 7.8 | 7.7 | 7.8 | 7.4 | 7.1 |

Table 15: Performance of different baselines with and without noise control in the Maze2D environments. The number of generated samples is 1M. The noise control technique can consistently improve the performance of different data augmentation methods.

| | SynthER w/ Noise control | SynthER w/o Noise control | ATraDiff w/ Noise control | ATraDiff w/o Noise control |
|---|---|---|---|---|
| maze2d-umaze | 2.3 | 1.7 | 3.1 | 2.8 |
| maze2d-medium | 0.6 | 0.3 | 1.6 | 1.2 |
| maze2d-large | 7.3 | 6.3 | 8.1 | 5.2 |

## C.5 QUANTITATIVE EVALUATION OF THE GENERATED SAMPLES

In this section, we conduct a quantitative evaluation of the generated samples of RTDiff. To measure the fidelity of the generated samples, we follow the previous works using two statistics: Marginal: Mean Kolmogorov-Smirnov (Massey Jr., 1951) and Correlation: Mean Correlation Similarity (Fieller et al., 1957). To measure the model error of the generated samples, we calculate the normalized error of the synthesized states and the real states after transition, which is $(T(s, a) - s')^2$ for a transition $(s, a, s')$. The results are presented in Tables 16 and 17.

As expected, the results show that RTDiff does not aim to generate more realistic trajectories, but rather to produce more diverse samples that lie outside the distribution, thereby benefiting the RL performance. This is because RTDiff generates adaptive, longer trajectories compared with other baselines, attributed to our proposed OOD detector and reverse synthesis model.

Therefore, while fidelity is an important factor in assessing data generation in general, our focus in this paper is more on the "usefulness" of the generated data, specifically how it improves RL performance.

Table 16: Performance comparison across different datasets for RTDiff, SynthER, and ATraDiff in terms of Marginal and Correlation metrics.

| Dataset | RTDiff | | SynthER | | ATraDiff | |
|---|---|---|---|---|---|---|
| | Marginal ↑ | Correlation ↑ | Marginal ↑ | Correlation ↑ | Marginal ↑ | Correlation ↑ |
| hopper-medium | 0.932 | 0.983 | 0.985 | 0.998 | 0.967 | 0.994 |
| hopper-medexp | 0.953 | 0.989 | 0.958 | 0.992 | 0.963 | 0.994 |
| hopper-expert | 0.941 | 0.985 | 0.934 | 0.982 | 0.953 | 0.991 |

## C.6 COMPARISON WITH VANILLA DIFFUSION MODEL

To clarify that vanilla diffusion models alone cannot match the performance of our RTDiff, we conducted an additional ablation study. This study involved using diffusion models to generate reverse trajectories without incorporating the other components of our method. The results, presented in Table 18, show that naïve diffusion models significantly underperform compared to our RTDiff.

Table 17: Model errors for RTDiff, SynthER, and ATraDiff across different Maze2D environments.

|  | RTDiff | SynthER | ATraDiff |
|---|---|---|---|
| maze2d-umaze | 0.05 | 0.02 | 0.03 |
| maze2d-medium | 0.06 | 0.03 | 0.03 |
| maze2d-large | 0.11 | 0.07 | 0.08 |

Table 18: Overall results of D4RL Maze2D environments with vanilla diffusion models. **The number denotes the performance increase by the data augmentation method compared to the original result.**

| Environment | Data Type | IQL (Kostrikov et al., 2022) | | | TD3+BC (Fujimoto & Gu, 2021) | | |
|---|---|---|---|---|---|---|---|
| | | **RTDiff** | **DM(vanilla)** | **ROMI** | **RTDiff** | **DM(vanilla)** | **ROMI** |
| maze2d | umaze | $\mathbf{8.3^{\pm 3.5}}$ | $4.3^{\pm 2.6}$ | $5.4^{\pm 2.4}$ | $\mathbf{10.2^{\pm 4.7}}$ | $9.3^{\pm 4.4}$ | $9.6^{\pm 3.3}$ |
| | medium | $\mathbf{3.3^{\pm 2.7}}$ | $2.3^{\pm 1.6}$ | $2.1^{\pm 1.8}$ | $\mathbf{9.8^{\pm 2.5}}$ | $8.9^{\pm 1.4}$ | $9.4^{\pm 1.7}$ |
| | large | $\mathbf{14.3^{\pm 3.3}}$ | $9.0^{\pm 3.1}$ | $8.1^{\pm 2.2}$ | $\mathbf{7.7^{\pm 4.5}}$ | $4.3^{\pm 2.6}$ | $3.5^{\pm 3.1}$ |

Furthermore, when comparing the performance of vanilla diffusion models with ROMI, the improvements, if any, are inconsistent and relatively minor. Therefore, the results validate our contribution, demonstrating that it extends beyond the straightforward application of diffusion models.

# D    ADDITIONAL VISUALIZATIONS

In this section, we show some additional visualization results in the D4RL (Fu et al., 2020) Maze2D-umaze environment. We collect offline data consisting of trajectories going from green ball to red ball and use it to train the data synthesizer. The *normal* synthesis will generate trajectories entering the obstacle area, while *reverse* synthesis can avoid this problem.

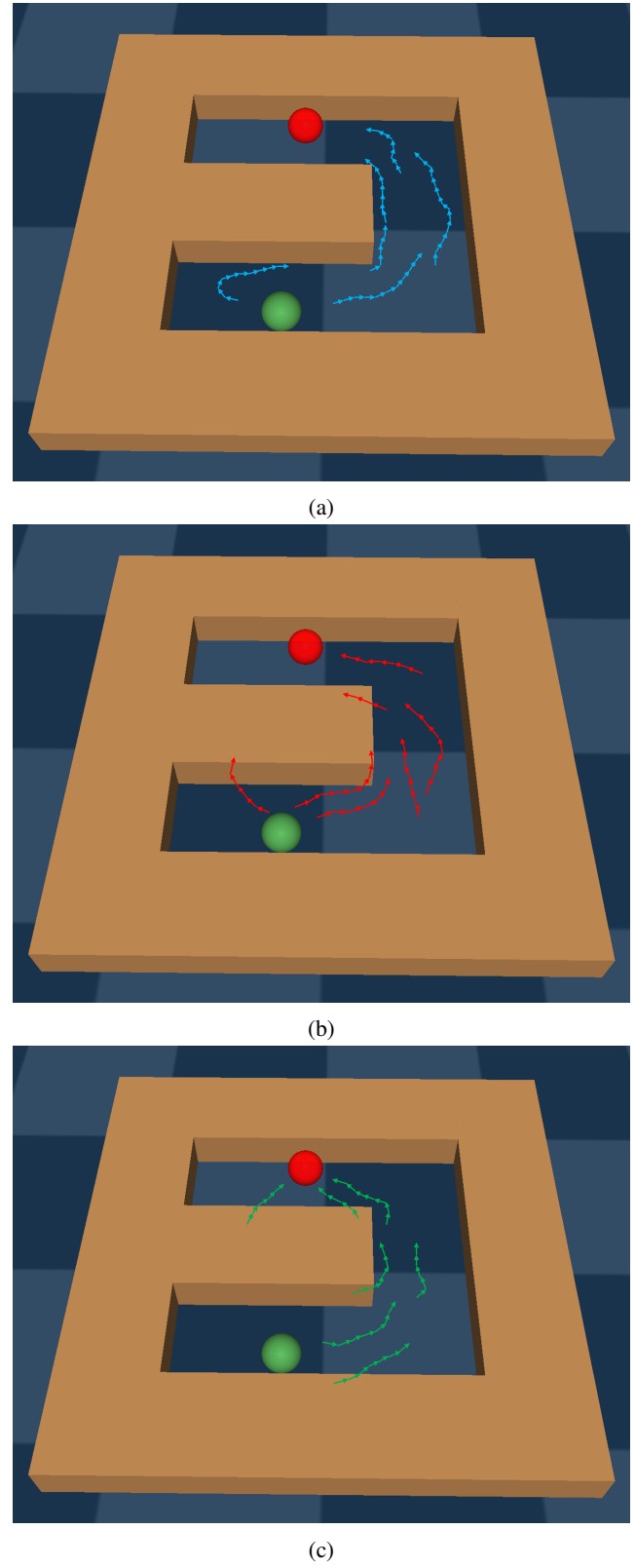

(a)

(b)

(c)

Figure 3: **Top**: The trajectory in the offline dataset. **Middle**: The trajectory of the offline RL agent with *normal* synthesis. **Bottom**: The trajectory of the offline RL agent with *reverse* synthesis.

