# OpenReview forum: "RTDiff: Reverse Trajectory Synthesis via Diffusion for Offline Reinforcement Learning"
_ICLR.cc/2025/Conference — ICLR 2025 Poster_

### Official Review · Reviewer_exWn · 2024-10-24

**Soundness:** 2
**Presentation:** 2
**Contribution:** 2
**Rating:** 5
**Confidence:** 3

**Summary:**

Traditional offline reinforcement learning methods often introduce conservative biases to limit exploration to familiar regions, but this can restrict an agent's ability to generalize. While recent approaches use generative models to expand offline datasets, they can overestimate synthesized data, particularly when it includes out-of-distribution samples. RTDiff is introduced to address this, which is a diffusion-based data augmentation technique that creates trajectories in reverse, moving from unknown to known states. This reverse approach reduces the risk of overestimation by ensuring the agent avoids planning through unfamiliar regions. It also supports generating longer trajectories, utilizing diffusion models effectively while maintaining reliability. RTDiff further optimizes the process with flexible trajectory length control and noise management to improve generation efficiency.

**Strengths:**

The reverse generation method, which has been empirically verified, effectively reduces the augmentation of data in risky regions. This concept is intuitively illustrated in the diagram.

**Weaknesses:**

1.Previous research [1] introduced a reverse data generation approach using a transition model. In the current work, the vanilla model is replaced with a diffusion model, yet the fundamental concept remains unchanged, limiting the overall contribution.

2.The explanation relies on an intuitive diagram, but it would be more effective to demonstrate several specific cases. Identifying which states are risky.  For example, some states are prone to be overestimated and easily generated by a forward model, but the reverse model effectively avoids generating them.

3.Minor errors are present, such as in line 156, where "dat" should be corrected to "data."

4.Environments based on visual data should be included in the analysis.

[1] Wang, Jianhao, et al. Offline reinforcement learning with reverse model-based imagination.

**Questions:**

Please see weaknesses.

---

> ### Author Response · Authors · 2024-11-25
> **Response to Reviewer exWn**
>
> Thanks for your insightful and inspiring comments! We provide the following clarifications in response to your concerns.
> 1. **W1: Comparison with ROMI**
> - We thank the reviewer for highlighting this related work. We have updated the paper to include a comparison of this work in Appendix C.5.
> We want to emphasize that our approach is significantly different from ROMI. First, ROMI focuses on learning the environment's dynamics and employing a reverse rollout strategy based on reverse dynamics. In contrast, our method directly generates trajectories using diffusion models, which avoids the inaccuracies inherent in dynamics learning and rollout strategies, resulting in higher-quality generated trajectories.
> - Additionally, our work introduces a novel strategy to fully leverage the diffusion model's ability to generate reliable long-length trajectories. Specifically, we incorporate an OOD detector to control the generation length. This detector ensures that the generated trajectories extend beyond the offline data distribution, introducing new environmental information to enhance offline RL algorithms. Simultaneously, it prevents trajectories from straying too far from the distribution, maintaining reliability and utility.
> - Furthermore, we develop a noise control strategy that can be applied to any diffusion-based data augmentation method. This technique is particularly effective when the number of samples is small and can be utilized with any diffusion-based data augmentation approach for RL.
> - Overall, our work introduces several distinct techniques compared to ROMI. To empirically demonstrate the advantages of RTDiff, we include ROMI as a baseline in our experiments. The results clearly show that RTDiff outperforms ROMI.
> | | IQL+RTDiff| IQL+ROMI| TD3BC+RTDiff| TD3BC+ROMI|
> | ------ | ------------------ | ---- | ---- | ---- |
> |maze2d-umaze| 8.3 |5.4|10.2|9.6|
> |maze2d-medium | 3.3 |2.1|9.8|9.4|
> |maze2d-large | 14.3 | 8.1|7.7|3.5|
> 2. **W2: Visualization of the experiments**
> - Thanks for the valuable comments. To better illustrate the generated trajectories in real environments used in the experiments, we have presented a new visualization of generated trajectories in the Maze2D environment in Appendix D of the updated version of the paper.
> 3. **W3: Typos**
> - Thanks for pointing this out. We have fixed this issue in the updated version of the paper.
> 4. **Analysis of visual environments**
> - First, we would like to clarify that we have already included the experimental results in the visual reinforcement learning environments in Section 5.3 of our original paper.
> - Here we provide an additional analysis in the Meta-world environment, similar to that presented in Section 6.. We choose Coffee Push as the example task. The results show that even in the visual environments, RTDiff still generates fewer risky transitions and more useful transitions.
>
> | | In2Out| Out2In | In2In | Out2Out |
> | ------ | ------ | ---- | ------ | --- |
> |RTDiff| 1.3% | 17.5% | 56.8% | 24.4% |
> |Forward (w/o OOD) | 16.0% | 3.9% | 68.2% | 11.9% |
> |Forward (w/ OOD) | 19.4% | 5.6% | 57.2% | 17.8% |

---

> > ### Comment · Reviewer_exWn · 2024-11-26
> >
> > My main concern is still the contribution, as the idea of employing a reverse rollout has been introduced previously, thus limiting this paper to decreasing inaccuracies with diffusion, which largely reduces the novelty. Hence, I maintain the score.

---

> > > ### Author Response · Authors · 2024-11-27
> > > **Response to Reviewer exWn**
> > >
> > > Thank you so much for your response and additional comments on the paper. We are very glad to address your remaining concern regarding our contributions in comparison to ROMI.
> > >
> > > First, we would like to highlight that the use of diffusion models for data augmentation in RL is a relatively emerging and underexplored area. We are the first work that introduces diffusion models for reserve trajectory synthesis. We believe this represents a significant contribution that should not be undervalued simply because of ROMI – as clarified in our previous response, ROMI still belongs to the traditional model-based methods.
> > >
> > > Second, we would like to emphasize that our motivation for applying diffusion models to trajectory generation extends beyond merely reducing inaccuracies compared to ROMI. Diffusion models possess several key advantages that we aim to leverage:
> > > - Diffusion models are particularly effective at generating **longer**, more accurate trajectories compared to traditional model-based methods like ROMI. Longer trajectories provide richer and more consistent information for offline RL agents, leading to improved performance. Our approach includes an out-of-distribution (OOD) detector that **adaptively determines trajectory length**, further enhancing the utility of long trajectories generated by diffusion models.
> > > - As a traditional model-based method, ROMI relies on **separate learned dynamics models and rollout strategies, introducing additional complexity**. Unlike ROMI and model-based methods in general, diffusion models naturally integrate these processes, mitigating such challenges. Moreover, diffusion models allow for innovative techniques, such as the noise control mechanism proposed in our work.
> > > - We believe our work establishes a more general and flexible framework for reverse trajectory synthesis, creating opportunities to leverage advancements in diffusion-based data generation. For example, exploring guidance-based diffusion generation to produce more tailored trajectories could be an interesting direction for future research.
> > >
> > > Third, our work also introduces key techniques to optimize diffusion-based trajectory generation: 1) **OOD Detector for Length Control:** This mechanism is vital for generating trajectories that extend beyond the offline data distribution, introducing novel environmental information while ensuring reliability by preventing trajectories from deviating excessively. 2) **Noise Control Strategy:** This generalizable technique enhances the diversity of generated trajectories, particularly when sample size is limited, and is applicable to any diffusion-based trajectory generation approach.
> > >
> > > Finally, **to clarify that vanilla diffusion models alone cannot match the performance of our RTDiff, we conducted an additional ablation study**. This study involved using diffusion models to generate reverse trajectories *without* incorporating the other components of our method. The results, presented below, show that naïve diffusion models significantly underperform compared to our RTDiff. Furthermore, when comparing the performance of vanilla diffusion models with ROMI, the improvements, if any, are inconsistent and relatively minor.
> > > Therefore, the results validate our contribution, demonstrating that **it extends beyond the straightforward application of diffusion models**.
> > >
> > > | | IQL+RTDiff| IQL+ DM(vanilla)|IQL+ROMI | TD3BC+RTDiff| TD3BC+DM(vanilla)| TD3BC+ROMI|
> > > | ------ | ------------------ | ---- | ---- | ---- | ---- | ---- |
> > > |maze2d-umaze| 8.3 |4.3|5.4|10.2|9.3|9.6|
> > > |maze2d-medium | 3.3 |2.3|2.1|9.8|8.9|9.4|
> > > |maze2d-large | 14.3 | 9.0|8.1|7.7|4.3|3.5|
> > >
> > > We hope our response has addressed your remaining concerns. If you have any further comments, we are more than happy to discuss them. We will incorporate all your comments and suggestions into our revision, which are invaluable for improving the quality of our work.

---

> > > ### Author Response · Authors · 2024-12-01
> > > **Further response to Reviewer exWn**
> > >
> > > Dear Reviewer exWn,
> > > Thank you for taking the time to review our manuscript and for your thoughtful feedback. We have updated the PDF to include the additional results requested in your previous response. We hope these results, along with our clarifications, address your concerns and provide further support for a re-evaluation of the paper.
> > > If you have any further questions or require additional clarification, we would be more than happy to address them.

---

> > > ### Author Response · Authors · 2024-12-02
> > > **Reply to Reviewer exWn**
> > >
> > > As the discussion period will come to an end in less than 24 hours, we would like to send you a reminder about our responses as above to solve your concerns. Please check whether your concerns have been addressed. We are sincerely looking forward to hearing from you, and are always happy to have more discussions with you!

---

> > > > ### Comment · Reviewer_exWn · 2024-12-03
> > > >
> > > > Thank you for your response. I appreciate your efforts in clarifying the manuscript. However, I believe that the additional components of your method represent a natural progression in applying the diffusion model for data generation, inspired by the ROMI framework. Additionally, it's important to consider that using a diffusion model for trajectory generation can lead to increased time and computational resource costs.

---

> > > > > ### Author Response · Authors · 2024-12-04
> > > > > **Reply to Reviewer exWn**
> > > > >
> > > > > Thank you for your valuable comments and your appreciation of our work. However, we respectfully disagree with the opinion that the additional components in our work represent a natural progression from ROMI in applying diffusion.
> > > > >
> > > > > First, we would like to reiterate that our work is substantially different from traditional model-based methods like ROMI and demonstrates notable strengths, as detailed in our previous responses, validated through our experimental comparison, and acknowledged by Reviewers **LuMS**, **FUCB**, and **NMzW**.
> > > > >
> > > > > Second, we would like to emphasize that the crucial components, such as generation length control, leverage the unique properties of diffusion models, making our approach distinct from a straightforward extension of ROMI. Additionally, to the best of our knowledge, we are **the first to utilize an out-of-distribution (OOD) detector to control the generation length for reinforcement learning data augmentation**. **It is a non-trivial insight to introduce the OOD detector for our reverse synthesis model, as forward synthesis methods cannot benefit from the OOD detector as empirically validated**.
> > > > >
> > > > > Furthermore, we would like to kindly point out the promising value and recognition of our work within the community by referring to publications of a similar nature. For example, [1] (one of our forward synthesis baselines) trained a diffusion model on an offline dataset to generate additional transitions. Despite sharing similarities with earlier model-based data augmentation methods, [1] was accepted at NeurIPS 2023.
> > > > >
> > > > > Regarding the concern about increased computational cost due to the diffusion model, we would like to clarify that, as mentioned in Section 4.1 of our paper (Line 237), we use a **lightweight architecture for the diffusion model**, which differs from typical diffusion architectures used for image generation and requires significantly fewer resources. Additionally, we want to emphasize that, as demonstrated by previous research [1,2,3], the primary bottleneck in reinforcement learning often lies in interacting with the environment, leading researchers to prioritize the number of samples used. Therefore, we respectfully suggest that the criticism regarding running time may be somewhat misplaced.
> > > > >
> > > > >
> > > > > [1] Synthetic Experience Replay. NeurIPS, 2023.
> > > > >
> > > > > [2] Diffusion Model is an Effective Planner and Data Synthesizer for Multi-Task Reinforcement Learning. NeurIPS, 2023.
> > > > >
> > > > > [3] Accelerating Online Reinforcement Learning with Imaginary Trajectories. ICML, 2024.

---

### Official Review · Reviewer_NMzW · 2024-11-01

**Soundness:** 3
**Presentation:** 3
**Contribution:** 3
**Rating:** 6
**Confidence:** 4

**Summary:**

This paper introduces a novel diffusion-based data augmentation method, RTDiff, for offline reinforcement learning. First, RTDiff mitigates the data distribution shift issue present in previous data augmentation methods by generating reverse trajectories instead of forward ones. Second, RTDiff trains an out-of-distribution (OOD) detector to truncate the OOD segments of generated trajectories, further enhancing sample authenticity. Finally, the authors propose a new noisy control method to improve sample generation efficiency. Experimental results validate the effectiveness and efficiency of RTDiff and different components across both vector-based and pixel-based tasks.

**Strengths:**

1. This paper is well-written and easy to follow.
2. RTDiff introduces reverse trajectory generation, an OOD detector, and a noisy control method to achieve efficient and reliable sample generation. These approaches are intuitively sound.
3. The authors conduct extensive benchmark and ablation experiments. Experimental results demonstrate that RTDiff outperforms previous baselines on both vector-based and pixel-based tasks, and each component—reverse trajectory generation, the OOD detector, and noisy control—exhibits its effectiveness.

**Weaknesses:**

1. Reverse trajectory generation is not new to offline RL, and the paper lacks a discussion and experimental comparison of prior works, such as [1].
2. The paper lacks a clear and well-reasoned explanation of the issues with previous data augmentation methods. In lines 88-90, this paper claims that previous data augmentation methods suffer from the data distribution shift, potentially leading to value overestimation. However, since all of these works use offline RL algorithms, I think data distribution shift is not the key issue. On the contrary, data augmentation is only effective when the generative model can produce samples that differ from the training data.
3. Generated data fidelity is a more critical factor. The paper lacks a quantitative evaluation of the fidelity of generated samples.

[1] Offline Reinforcement Learning with Reverse Model-based Imagination. NeurIPS, 2021.

**Questions:**

See weaknesses.

---

> ### Author Response · Authors · 2024-11-25
> **Response to Reviewer NMzW (part 1/2)**
>
> Thanks for your insightful and inspiring comments! We provide the following clarifications in response to your concerns.
> 1. **W1: Comparison with ROMI**
> - We thank the reviewer for highlighting this related work. We have updated the paper to include a comparison of this work in Appendix C.5.
> We want to emphasize that our approach is significantly different from ROMI. First, ROMI focuses on learning the environment's dynamics and employing a reverse rollout strategy based on reverse dynamics. In contrast, our method directly generates trajectories using diffusion models, which avoids the inaccuracies inherent in dynamics learning and rollout strategies, resulting in higher-quality generated trajectories.
> - Additionally, our work introduces a novel strategy to fully leverage the diffusion model's ability to generate reliable long-length trajectories. Specifically, we incorporate an OOD detector to control the generation length. This detector ensures that the generated trajectories extend beyond the offline data distribution, introducing new environmental information to enhance offline RL algorithms. Simultaneously, it prevents trajectories from straying too far from the distribution, maintaining reliability and utility.
> - Furthermore, we develop a noise control strategy that can be applied to any diffusion-based data augmentation method. This technique is particularly effective when the number of samples is small and can be utilized with any diffusion-based data augmentation approach for RL.
> - Overall, our work introduces several distinct techniques compared to ROMI. To empirically demonstrate the advantages of RTDiff, we include ROMI as a baseline in our experiments. The results clearly show that RTDiff outperforms ROMI.
> | | IQL+RTDiff| IQL+ROMI| TD3BC+RTDiff| TD3BC+ROMI|
> | ------ | ------------------ | ---- | ---- | ---- |
> |maze2d-umaze| 8.3 |5.4|10.2|9.6|
> |maze2d-medium | 3.3 |2.1|9.8|9.4|
> |maze2d-large | 14.3 | 8.1|7.7|3.5|
> 2. **W2: Explanation of the idea**
> - We want to clarify that *the core idea of this work is not to directly reduce the ratio of generated data that falls outside the offline data distribution but rather to mitigate the issues caused by such trajectories*. The problem with data distribution arises from the overestimation of transitions in the fake trajectories generated during data augmentation. These overestimated transitions can directly impact the strategies learned by the agents, leading to degraded performance.
> - Our approach addresses this issue by performing synthesis in reverse order. This ensures that even if some trajectories deviate from the offline data distribution, the overestimation of such transitions does not directly harm performance.
> - As analyzed in Section 6, we categorize transitions into four types, where only the In2In transitions are strictly within the offline data distribution. Among the other three categories, the In2Out transitions are particularly detrimental to performance. Our goal is not to eliminate the other three categories entirely or ensure that all generated transitions are in-distribution. Instead, we focus on reducing the number of In2Out transitions.
> - As demonstrated in Table 7, reverse synthesis can even result in fewer In2In transitions, which are less useful for data augmentation. This finding further validates the effectiveness of our approach.

---

> ### Author Response · Authors · 2024-11-25
> **Response to Reviewer NMzW (part 2/2)**
>
> 3. **W3: Fidelity of the generated trajectories**
> Thanks for your valuable comments. We have added a section in Appendix C.6 of the updated version of the paper to include the quantitative evaluation of generated trajectories.
> Below we show the fidelity of the generated trajectories. To measure the fidelity of the generated samples, we follow the previous works using two statistics: Marginal: Mean Kolmogorov-Smirnov [Ref1] and Correlation: Mean Correlation Similarity [Ref2]. As expected, the results show that RTDiff does not aim to generate more realistic trajectories, but rather to produce more diverse samples that lie outside the distribution, thereby benefiting the RL performance. This is because RTDiff generates adaptive, longer trajectories compared with other baselines, attributed to our proposed OOD detector and reverse synthesis model.
> Therefore, we would like to further emphasize that, while we agree with the reviewer that fidelity is an important factor in assessing data generation in general, our focus here is more on the "usefulness" of the generated data, specifically how it improves RL performance.
>
> | Dataset            | RTDiff Marginal $\uparrow$ | RTDiff Correlation $\uparrow$ | SynthER Marginal $\uparrow$ | SynthER Correlation $\uparrow$ | ATraDiff Marginal $\uparrow$ | ATraDiff Correlation $\uparrow$ |
> |--------------------|------------------|---------------------|---------------|------------------|---------------|------------------|
> | hopper-medium | 0.932 | 0.983 | 0.985        | 0.998           | 0.967         | 0.994            |
> | hopper-medexp | 0.953 | 0.989 |  0.958            | 0.992               | 0.963     | 0.994        |
> | hopper-expert      | 0.941 | 0.985 |  0.934            | 0.982               | 0.953     | 0.991        |
>
> [Ref1] The kolmogorov-smirnov test for goodness of fit. Frank J. Massey Jr. 1951.
>
> [Ref2] Tests for rank correlation coefficients. E. C. Fieller, et al. 1957.

---

> > ### Comment · Reviewer_NMzW · 2024-11-26
> >
> > Thank you for your detailed response, which thoroughly addresses my concerns. I am willing to raise my score to 6.

---

> > > ### Author Response · Authors · 2024-11-26
> > > **Reply to Reviewer NMzW**
> > >
> > > Thank you so much for your response and for increasing the score! We are glad that our rebuttal has addressed the concerns raised.
> > > If you have any further questions, we are more than happy to discuss them. We will incorporate all your comments and suggestions into our revision, which are invaluable for improving the quality of our work.

---

### Official Review · Reviewer_FUCB · 2024-11-03

**Soundness:** 3
**Presentation:** 3
**Contribution:** 2
**Rating:** 6
**Confidence:** 4

**Summary:**

This paper proposes RTDiff, a novel diffusion-based data augmentation technique that synthesizes trajectories in a reverse direction. Such reverse generation naturally mitigates the risk of overestimation by ensuring that the agent avoids planning through unknown states. RTDiff also introduces some other tricks including flexible trajectory control and noise management to improve sythesis quality. Emprirical results show the advantage of reverse generation over forward generation.

**Strengths:**

1. The article is well-written, with clear expression and logic, effectively reflecting the main argument.
2. The experiments are very comprehensive. The authors validate the effectiveness of their method across a series of tasks, including both proprioceptive observations and visual observations.

**Weaknesses:**

1. I believe the authors miss a key related work (called ROMI [1]), which is the first to propose using reverse trajectory generation in the field of offline reinforcement learning. The motivation described for reverse trajectory generation in these two work is also very similar. Therefore, considering that this paper is neither the first to propose the use of reverse trajectory generation in offline RL nor the first to use diffusion for data augmentation, I would say the novelty of this paper is quite limited.
2. I think this paper lacks comparisons with model-based offline RL methods in the experimental section. According to my understanding, using a diffusion model for data augmentation essentially falls under the same category as previous model-based offline RL methods. For instance, MOPO [2] essentially generates a batch of synthetic samples to supplement the original samples. Therefore, the authors should compare some model-based offline RL methods, such as MOPO, RAMBO [3], etc.

[1] Wang et al. "Offline Reinforcement Learning with Reverse Model-based Imagination" (NeurIPS'21)
[2] Yu et al. "MOPO: Model-based Offline Policy Optimization" (NeurIPS'20)
[3] Rigter et al. "RAMBO-RL: Robust Adversarial Model-Based Offline Reinforcement Learning" (NeurIPS'22)

**Questions:**

1. Can the authors discuss in detail the differences between this paper and previous similar works (like ROMI)?
2. This method is built on IQL, TD3BC, and CQL. Have there been any adjustments to the hyperparameters of these methods after using data augmentation?
3. Is it possible to conduct a quantitative evaluation of the quality of the generated trajectories? For example, assessing the model error of the generated trajectories, etc.

---

> ### Author Response · Authors · 2024-11-25
> **Response to Reviewer FUCB (part 1/2)**
>
> Thanks for your insightful and inspiring comments! We provide the following clarifications in response to your concerns.
>
> In particular, we discuss in detail how our approach significantly differs from ROMI and model-based RL methods, providing empirical comparisons that validate our approach's superior performance. Additionally, we would like to emphasize that the use of diffusion models for data augmentation in RL is a relatively emerging area. Prior works such as SynthER and ATraDiff also focus on using diffusion models for data augmentation, with their novelty stemming from the specific techniques used to exploit diffusion models. Within this context, we believe the novelty of our work should not be diminished solely because we are not the first to use diffusion for data augmentation.
>
> 1. **W1/Q1: Comparison with ROMI**
> - We thank the reviewer for highlighting this related work. We have updated the paper to include a comparison of this work in Appendix C.5.
> We want to emphasize that our approach is significantly different from ROMI. First, ROMI focuses on learning the environment's dynamics and employing a reverse rollout strategy based on reverse dynamics. In contrast, our method directly generates trajectories using diffusion models, which avoids the inaccuracies inherent in dynamics learning and rollout strategies, resulting in higher-quality generated trajectories.
> - Additionally, our work introduces a novel strategy to fully leverage the diffusion model's ability to generate reliable long-length trajectories. Specifically, we incorporate an OOD detector to control the generation length. This detector ensures that the generated trajectories extend beyond the offline data distribution, introducing new environmental information to enhance offline RL algorithms. Simultaneously, it prevents trajectories from straying too far from the distribution, maintaining reliability and utility.
> - Furthermore, we develop a noise control strategy that can be applied to any diffusion-based data augmentation method. This technique is particularly effective when the number of samples is small and can be utilized with any diffusion-based data augmentation approach for RL.
> - Overall, our work introduces several distinct techniques compared to ROMI. To empirically demonstrate the advantages of RTDiff, we include ROMI as a baseline in our experiments. The results clearly show that RTDiff outperforms ROMI.
> | | IQL+RTDiff| IQL+ROMI| TD3BC+RTDiff| TD3BC+ROMI|
> | ------ | ------------------ | ---- | ---- | ---- |
> |maze2d-umaze| 8.3 |5.4|10.2|9.6|
> |maze2d-medium | 3.3 |2.1|9.8|9.4|
> |maze2d-large | 14.3 | 8.1|7.7|3.5|
> 2. **W2: Comparisons with model-based RL**
> - We thank the reviewer for highlighting that model-based RL methods can also serve as strategies for offline data augmentation. However, the generation quality of model-based methods heavily depends on both the accuracy of the learned dynamics and the rolling-out strategy used to produce trajectories. These factors can make it challenging to achieve reliable data augmentation for RL. It is worth noting that ROMI is an example of a model-based data augmentation method.
> - Our approach, in contrast, incorporates several novel components specifically designed to improve the quality of generated data, such as the OOD detector and noise control mechanisms.
> - To provide an empirical comparison between our method and model-based RL approaches, we have included an additional experiment in Appendix C.5 of the updated version of the paper, comparing our method to MOPO as suggested by the reviewer. The results below demonstrate that our method outperforms model-based methods in terms of performance, further highlighting its effectiveness.
> | | IQL+RTDiff| IQL+MOPO| TD3BC+RTDiff| TD3BC+MOPO|
> | ------ | ------------------ | ---- | ---- | ---- |
> |maze2d-umaze| 8.3 | 5.1 | 10.2 | 9.6 |
> |maze2d-medium | 3.3 | 1.7 | 9.8 | 8.8 |
> |maze2d-large | 14.3 | 5.9 | 7.7 | 2.6 |
> 3. **Q2: RL methods hyperparameters**
> - After the data augmentation, we did not adjust any hyperparameters of the RL methods.

---

> ### Author Response · Authors · 2024-11-25
> **Response to Reviewer FUCB (part 2/2)**
>
> 4. **Q3: Quantitative evaluation of generated trajectories**
> - Thanks for your valuable comments. We have included this in Appendix C.6 of the updated version of the paper, where we present the quantitative evaluation of generated trajectories.
> - Below we show the average model error of the generated trajectories. To measure the model error of the generated samples, we calculate the normalized error between the synthesized states and the real states after transition, which is defined as $(T(s,a) - s')^2$ for a transition $(s, a, s')$. As expected, the results show that RTDiff does not aim to generate more realistic trajectories, but rather to produce more diverse samples that lie outside the distribution, thereby benefiting the RL performance. This is because RTDiff generates adaptive, longer trajectories compared with other baselines, attributed to our proposed OOD detector and reverse synthesis model.
> - Therefore, we would like to further emphasize that, while we agree that the quality is an important factor in assessing data generation, our focus here is more on the "usefulness" of the generated data, specifically how it improves RL performance.
> | | RTDiff | SynthER | ATraDiff |
> | ------ | ------------------ | ---- | ---- |
> |maze2d-umaze| 0.05 | 0.02 | 0.03 |
> |maze2d-medium | 0.06 | 0.03 | 0.03 |
> |maze2d-large | 0.11 | 0.07 | 0.08 |

---

> ### Comment · Reviewer_FUCB · 2024-11-25
>
> Thanks for the authors' detailed reply. I have no further questions and I'm willing to raise my score.

---

> > ### Author Response · Authors · 2024-11-26
> > **Reply to Reviewer FUCB**
> >
> > Thank you so much for your response and for increasing the score! We are glad that our rebuttal has addressed the concerns raised.
> > If you have any further questions, we are more than happy to discuss them. We will incorporate all your comments and suggestions into our revision, which are invaluable for improving the quality of our work.

---

### Official Review · Reviewer_LuMS · 2024-11-03

**Soundness:** 3
**Presentation:** 3
**Contribution:** 3
**Rating:** 6
**Confidence:** 4

**Summary:**

To address the distribution shift issue in offline reinforcement learning, this paper proposes a novel diffusion-based data augmentation technique, namely RTDiff, where rather than generating forward trajectories, the reverse one is synthesized, which is also the paper's main contribution. Furthermore, the performance of RTDiff is enhanced with the introduction of trajectory length control and noise management. Experimental results show the effectiveness of the proposed approach.

**Strengths:**

The ideas in this paper are interesting and novel, where to my knowledge, this is the first work that utilizes the concept of generating the reverse trajectories to address the distribution shift issue. The paper is clearly written and well-motivated.  The effectiveness of the proposed method is verified in various environments, and ablation studies are also conducted to validate the effectiveness of different components of the proposed method.

**Weaknesses:**

The main concern of the paper is whether the reverse synthesis can actually address the issue of distribution shift. As it is not state clearly whether the OOD detector is incorporated with other data augmentation baselines, e.g., SynthER and ATraDiff, it is not very certain whether the better performance achievement of RTDiff is due to the reverse synthesis or the using of the OOD detector. In Section 6, an analysis is conduct by using a simple illustrative environment to show why reverse synthesis avoids issues present in normal synthesis. However, on the one hand, it is better to directly use an environment used in the experiments for the analysis, on the other hand, it is confusing why the reverse synthesis generates trajectories that move from the dangerous area to the upper or lower areas while normal synthesis generates trajectories that start from the lower area and enter the middle dangerous area. Do these two approaches both start from a state in the offline data, and then generate the trajectories in different ways? If the OOD detector is used in the normal synthesis, can the dangerous areas also be avoided? Moreover, the theoretical contribution of the proposed is not very significant.
If the concerns can be addressed, I would like to raise my score.

**Questions:**

1. Are the OOD detector and noise management incorporated with other data augmentation baselines, e.g., SynthER and ATraDiff?

2.In Section 6, a very specific environment is adopted to show the advantages of reverse synthesis over normal synthesis, what is the reason/possible explanation that normal synthesis with OOD detector even produces significantly more In2Out than normal synthesis without OOD detector? (18.2%11.2)

3. For other questions, please see questions raised in Weaknesses.

---

> ### Author Response · Authors · 2024-11-25
> **Response to Reviewer LuMS (part 1/2)**
>
> Thanks for your insightful and inspiring comments! We provide the following clarifications in response to your concerns:
>
> 1. **Incorporating OOD detector with other baselines**
>
> - Thanks for the valuable question. We would like to clarify that the *OOD detector is particularly useful for our reverse synthesis model, where forward synthesis methods cannot benefit from the OOD detector*.
> - Specifically, in our method, the OOD detector is used to control the length of the generated trajectories. This is crucial because we aim to generate trajectories that extend beyond the offline data distribution to provide new information to the agent, while ensuring they do not deviate too far, which could reduce their usefulness and increase risk. The effectiveness of our OOD detector is demonstrated in Tables 4 and 12. Our results show that the OOD detector outperforms any fixed-length generation strategy. Additionally, we observed that setting the threshold too high or too low negatively impacts performance.
> - To better understand why reverse synthesis is useful, we want to further show that this OOD detector is not useful for other synthesis methods. First of all, SynthER only performs transition-level synthesis, which inherently does not need to control the generation length.  Also we want to emphasize that with forward synthesis like ATraDiff, generating trajectories going out of the offline data distribution is more risky, as transitions going from inside to outside may lead to performance degradation. To validate this, we show the results of combining ATraDiff with an OOD detector of different thresholds. From the results below, we found that those forward synthesis methods derive limited benefits from the OOD detector, which validated the unique effectiveness of reverse synthesis in our framework.
> | | RTDiff (Ours) | ATraDiff | ATraDiff with dis_m = 1.0 | ATraDiff with dis_m = 1.3 | ATraDiff with dis_m = 1.5 | ATraDiff with dis_m = 2.0
> | --- | ----- | ---- | ---- | ---- | ---- | ---- |
> |maze2d-umaze|12.3 | 7.1 | 7.4 |  7.0 | 6.3 | 6.0 |
> |maze2d-medium |8.3 |6.2 | 6.6 | 6.3 | 5.6 | 5.4 |
> |maze2d-large | 11.3 | 7.8 | 7.7 | 7.8 | 7.4 | 7.1|
> 2. **Incorporating noise control with other baselines**
> - Our proposed noise control is largely independent of the specific generation method and can be applied to any data augmentation approach using a diffusion-based framework. It demonstrates particularly effective when the number of examples is limited. Notably, this technique is also applicable to SynthER and ATraDiff, as demonstrated by the results shown below. We use SynthER and ATraDiff to generate both 1M data with and without the random generation. The results show that the noise control can consistently improve the performance.
> | | SynthER with Noise control| SynthER with random generation | ATraDiff with Noise control | ATraDiff with random generation |
> | ------ | ------------------ | ---- |  ---- | ---- |
> | maze2d-umaze | 2.3 | 1.7 | 3.1 | 2.8 |
> | maze2d-medium | 0.6 | 0.3| 1.6 | 1.2 |
> |maze2d-large | 7.3 | 6.3 | 8.1 | 5.2 |
> 3. **Explanation of illustration examples**
> - Thanks for the valuable comment. We first sincerely apologize for any confusion and would like to clarify that the reviewer may have misunderstood our illustration example. The results presented in Table 7 were generated in the Maze2D-large environment, not in the environment depicted in Figure 2.
> - The trajectories depicted in Figure 2 are intended to intuitively illustrate the behavioral differences between reverse synthesis and normal forward synthesis, highlighting the benefits of reverse synthesis. These trajectories are purely for illustrative purposes and were not generated by the models.
> - To better showcase the real generated trajectories in the environments investigated in this paper, we have included a new visualization of the generated trajectories in the Maze2D environment in Appendix D of the updated version of the paper.
> 4. **Why normal synthesis with OOD detector has more In2Out transitions?**
> - Thanks for the question. This is because the generation length in normal synthesis without an OOD detector is set to be a fixed length, which must be kept shorter than the average generation length with an OOD detector to avoid performance decrease from generating risky trajectories. In contrast, forward synthesis with an OOD detector allows for an increased average generation length, because the OOD detector can avoid generating too long trajectories. Therefore, forward synthesis with an OOD detector can generate some trajectories different from the offline data distribution. As demonstrated by the results in Table 2, forward synthesis with an OOD detector can also produce more transitions of Out2Out and significantly fewer In2In transitions which might be redundant.

---

> > ### Comment · Reviewer_LuMS · 2024-11-28
> >
> > Thanks for the authors' detailed response and the addition of comprehensive experimental results to further verify the effectiveness of the proposed method. My concerns have been addressed. I also read other reviews and the authors' corresponding responses. Other reviewers are concerned about the novelty of the proposed reverse trajectory generation approach compared to ROMI. In their rebuttal and the revision, the authors have explained the differences between ROMI and their approach. Also, experimental results show the superiority of RTDiff over ROMI. To me, the explanation and results are convincing, and the score has been raised.

---

> > > ### Author Response · Authors · 2024-11-28
> > > **Response to Reviewer LuMS**
> > >
> > > Thank you so much for your response and for increasing the score! We are glad that our rebuttal has addressed the concerns raised. If you have any further questions, we are more than happy to discuss them. We will incorporate all your comments and suggestions into our revision, which are invaluable for improving the quality of our work.

---

> ### Author Response · Authors · 2024-11-25
> **Response to Reviewer LuMS (part 2/2)**
>
> 5. **Theoretical contributions**
> - This paper primarily focuses on algorithmic development and empirical evaluation, rather than on theoretical analysis or formal proofs. Similar to previous works like ROMI, SynthER, and ATraDiff, our objective is to propose practical methodologies and validate their effectiveness through comprehensive experimental results. We leave the exploration of theoretical guarantees and formal proofs as interesting future work.

---

### Author Response · Authors · 2024-11-25
**General Response**

We are deeply grateful for the time and effort all the reviewers have invested in reviewing our manuscript. The reviewers’ constructive suggestions have been invaluable in enhancing the quality of our work. We particularly appreciate reviewers’ remarks acknowledging the strengths of our work:
- (LuMS) The idea of this paper is novel.
- (LuMS, FUCB, NMzW) The empirical results are comprehensive and the performance is good.

In response to the insightful questions raised, we have provided detailed clarifications in the direct responses. These elaborations aim to address the reviewers’ concerns comprehensively. We hope that the clarification and additional experimental results resonate with the reviewers’ expectations and adequately address the issues highlighted.

In particular, we have discussed the differences between RTDiff and ROMI, focusing on both technical distinctions and empirical comparisons. We hope this discussion effectively addresses the reviewers' questions.

In the updated version of the paper, we have added more experiments, including comparisons with additional baselines, quantitative evaluations of the generated samples, and more ablation studies in the appendix. The updated sections are highlighted in orange.

---

### Meta-Review · Area_Chair_jn7J · 2024-12-21

**Metareview:**

RTDiff proposes a novel diffusion-based approach for reverse trajectory synthesis in offline reinforcement learning, addressing the critical issue of distribution shift with mechanisms like an OOD detector and noise control. The authors provide extensive empirical evidence showcasing significant improvements over baselines, including ROMI, and demonstrate that the reverse synthesis strategy generates more reliable trajectories, enhancing agent performance. While concerns about the novelty relative to ROMI and computational efficiency were raised, the inclusion of diffusion models introduces a unique generative capability, and the provided ablation studies validate the contribution of each component. The paper represents a meaningful step forward in diffusion-based RL research and should be accepted.

**Additional Comments On Reviewer Discussion:**

The discussion centered on comparisons with ROMI, computational costs, and the novelty of the reverse synthesis approach. Critical reviewer exWn argued that the contribution was incremental and highlighted inefficiencies in diffusion-based generation, while others, including LuMS, FUCB, and NMzW, were convinced by the authors' added comparisons, ablations, and clarified novelty claims. The authors effectively differentiated RTDiff from ROMI by emphasizing its reliance on diffusion models and added mechanisms like OOD detectors and noise control, ultimately persuading most reviewers to support the paper. These additions and the strong empirical results outweighed the remaining concerns, leading to a recommendation for acceptance.

---

### Decision · Program_Chairs · 2025-01-22

Accept (Poster)